# PROMPTUS: REPRESENTING REAL-WORLD VIDEO AS STABLE DIFFUSION PROMPTS FOR VIDEO STREAMING

## ABSTRACT

With the exponential growth of video traffic, traditional video streaming systems are approaching their limits in compression efficiency and communication capacity. To further reduce bitrate while maintaining quality, we propose **Promptus**, a disruptive novel system that *streaming prompts instead of video content*, which represents real-world video frames with a series of "prompts" for delivery and employs Stable Diffusion to generate videos at the receiver. To ensure that the prompt representation is pixel-aligned with the original video, a gradient descent-based prompt fitting framework is proposed. Further, a low-rank decomposition-based bitrate control algorithm is introduced to achieve adaptive bitrate. For inter-frame compression, a temporal smoothing-based prompt interpolation algorithm is proposed. Evaluations across various video genres demonstrate that, compared to H.265, Promptus can achieve more than a 4x bandwidth reduction while preserving the same perceptual quality. On the other hand, at extremely low bitrates, Promptus can enhance the perceptual quality by 0.139 and 0.118 (in LPIPS) compared to VAE and H.265, respectively, and decreases the ratio of severely distorted frames by 89.3% and 91.7%. Our work opens up a new paradigm for efficient video communication. **Promptus will be open-sourced after publication.**

## 1    INTRODUCTION

With the rapid development of streaming applications (such as Youtube, Netflix and Disney+) , the traffic of network video has been continuously growing. To reduce traffic, video codecs represented by VP8 (Bankoski et al., 2011), VP9 (Mukherjee et al., 2015), H.264 (264, 2024) and H.265 (265, 2024) are widely used to compress videos. These codecs achieve compression by removing spatial and temporal redundancies. However, these redundancies are limited, so there is an upper bound on the compression ratio (subject to the Shannon limit (Shannon, 1948)). In order to further compress the video, non-redundant content in the video will be discarded, which will greatly degrade the video quality, such as causing blurring and blocking artifacts. In recent years, some deep learning-based codecs (Lu et al., 2019; Lin et al., 2020; Djelouah et al., 2019) and streaming frameworks (Zhou et al., 2022; Jiang et al., 2022; Sivaraman et al., 2024; Li et al., 2023) have been proposed to improve compression ratio, but they are limited either by performance or by generality.

With the popularity of generative AI, Stable Diffusion (Rombach et al., 2022; sd, 2024) has attracted extensive attention thanks to its powerful text-to-image generation capability. By pre-training on an internet-scale dataset LAION (2024), Stable Diffusion learns prior knowledge of nearly all human visual domains, and simultaneously learns the mapping from text to images. Therefore, Stable Diffusion can generate high-fidelity images based on a brief prompt composed of a few words.

**We are motivated by this question: is it possible for Stable Diffusion to replace the video codecs? During streaming, the sender streams prompts instead of streaming encoded videos and the receiver generates videos instead of decoding videos. In this way, the traffic of network video is reduced from the video scale to the text scale, greatly improving video communication efficiency.** In this paper, we propose Promptus, a system that represents real-world video frames with Stable Diffusion prompts, achieving ultra-low bitrate video streaming. To bring this vision to fruition, we address the following technical challenges:

First, how to ensure **pixel alignment** between the generated frames and the real frames. To invert a frame into a prompt, the most straightforward and powerful approach is to manually write a tex-

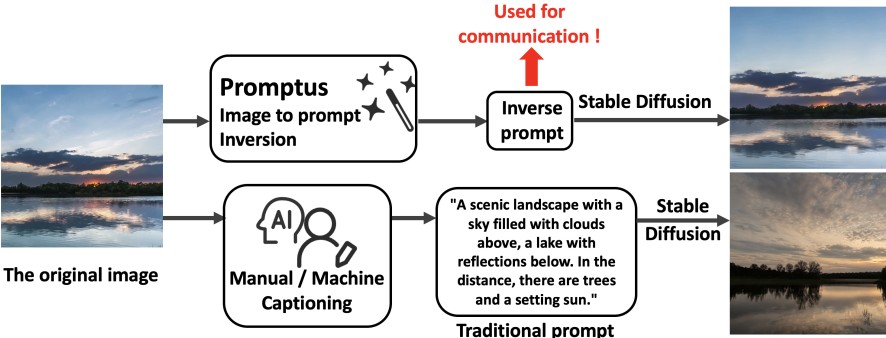

Figure 1: Promptus can invert a given image into a prompt. Based on this prompt, Stable Diffusion can generate an almost identical image to the original. In contrast, existing methods can only generate semantically similar images, while the differences at the pixel level are substantial. In this way, Promptus streams prompts instead of videos, significantly reducing bandwidth overhead.

tual description. However, the frame generated using this description can only guarantee semantic consistency with the original frame, while the differences at the pixel level are often substantial, as shown in Figure 1. To achieve pixel alignment, Promptus proposes a gradient descent-based prompt fitting framework. Specifically, the prompt is randomly initialized and then used to generate frame. The pixel-wise loss between the generated frame and the real frame is calculated. The partial derivative of the loss with respect to the prompt will be calculated and the prompt is iteratively optimized using gradient descent. To implement this framework, Promptus employs single-step denoising instead of iterative denoising to avoid higher-order derivatives. Second, Promptus uses embeddings as prompts instead of text to avoid non-differentiability. Third, Promptus uses a noisy previous frame instead of random noise to reduce latent space distance. Fourth, Promptus combines reconstruction and perceptual loss to enhance the perceptual quality of the generated frames.

Second, how to **control the bitrate** of the prompt. Since Promptus uses embeddings as prompts, which are matrices with fixed dimensions, the bitrate of the prompt cannot adapt to dynamic network bandwidth. To address this, Promptus proposes a low-rank bitrate control algorithm. Specifically, Promptus integrates the inverse process of low-rank decomposition into the gradient descent, directly fitting the decomposed prompt. The rank is used to control the trade-off between the quality and bitrate of the prompt. When the rank is higher, the representational capability of the prompt is better, and it can describe more details in the frame, but the data size is also larger. Therefore, Promptus adaptively selects the prompt rank based on the currently available bandwidth.

Third, how to perform **inter-frame compression** on prompts. Promptus inverts each frame into an independent prompt without considering the correlation across frames. Therefore, when streaming, Promptus needs to transmit prompts for each frame, resulting in the bitrate increasing linearly with the frame rate. To address this, Promptus adds a temporal smoothing regularization during prompt fitting, ensuring that temporally close frames are also sufficiently close in the prompt space. With this, Promptus only needs to sparsely transmit prompts for a few keyframes, while the prompts for the remaining frames can be approximated through linear interpolation of the keyframe prompts.

We evaluated Promptus on test videos from different domains. The results show that: First, Promptus achieves scalable bitrate, and its quality advantage over the baselines becomes more significant as the target bitrate decreases. Second, Promptus is general to different video domains, and the more complex the video content, the greater the quality advantage of Promptus compared to the baselines. Third, compared to H.265 (265, 2024), Promptus provides more than a 4x bandwidth reduction while preserving the same perceptual quality. On the other hand, at extremely low bitrates, Promptus can enhance the perceptual quality by 0.139 and 0.118 (in LPIPS (Zhang et al., 2018)) compared to VAE (Kingma & Welling, 2014; Rombach et al., 2022) and H.265, respectively, and decreases the ratio of severely distorted frames (whose LPIPS is higher than 0.32) by 89.3% and 91.7%.

The contributions of this paper are summarized as follows: (1) We propose Promptus, which, to the best of our knowledge, is the first attempt to replace video codecs with prompt inversion and also the first to use prompt streaming to replace video streaming. (2) We propose a gradient descent-based prompt fitting framework, achieving pixel-aligned prompt inversion. (3) We build a video streaming system based on prompts, significantly boosting video communication efficiency.

## 2 RELATED WORK AND BACKGROUND

### 2.1 VIDEO CODEC AND STREAMING

**Video codecs.** In network video traffic, most of the content is encoded using traditional codecs represented by VP8 (Bankoski et al., 2011), VP9 (Mukherjee et al., 2015), H.264 (264, 2024) and H.265 (265, 2024). These traditional video codecs achieve compression primarily by eliminating redundancy in the video. Specifically, the codecs exploit the correlation between adjacent pixel blocks to reduce spatial redundancy in intra-frame prediction. Besides, considering the similarity between consecutive video frames, the codecs employ inter-frame prediction techniques to eliminate temporal redundancy. Although these techniques achieve efficient compression, the compression ratio has an upper limit due to the finite amount of redundancy. With the development of deep learning, handcrafted modules in traditional codecs are being replaced by neural networks (Lu et al., 2019; Lin et al., 2020; Djelouah et al., 2019; Rippel et al., 2019; Li et al., 2024a; Sheng et al., 2024). Compared to handcrafted modules, these embedded neural networks can learn more complex and nonlinear motion patterns and intra-frame correlations, thereby achieving lower distortion at the same bitrate. However, since neural-embedded codecs still adhere to the traditional coding framework (aiming to fully remove redundancy), the improvement in compression ratio is limited.

**Neural-enhanced streaming.** In contrast to reducing redundancy, neural-enhanced streaming (Park et al., 2023; Zhou et al., 2022) actively discards most of the information (including non-redundant information), transmitting highly distorted low-bitrate videos. Then, at the receiver side, neural network-based post-processing algorithms (such as super-resolution) are used to restore high-fidelity details. Thanks to the powerful image restoration capabilities of neural networks, the lost details can be effectively recovered, thus ensuring video quality while achieving substantial compression. However, these post-processing neural networks rely on prior knowledge learned from training datasets. Due to the complex and diverse nature of real-world video, there exists domain gaps with the training set. The performance often degrades in unseen scenarios (Yang et al., 2021). A feasible solution is to fine-tune the post-processing neural networks for each video segment and send the fine-tuned neural networks along with the video (Yeo et al., 2018; 2022; 2020; Kim et al., 2020; Zhang et al., 2022). However, the transmission of the neural networks inevitably increases the overall bitrate.

**Generative streaming.** Compared to using neural networks for post-processing, generative streaming directly uses neural networks to generate videos to handle higher compression ratios (Jiang et al., 2022; Wang et al., 2021; Sivaraman et al., 2024; Li et al., 2023). For example, in the context of video conferencing, Face-vid2vid (Wang et al., 2021; Jiang et al., 2022) extracts facial keypoints in real-time at the sender side for transmission. At the receiver side, it generates dynamic facial videos using the keypoints and static facial images. The bitrate of facial keypoints is much lower than that of videos, thus compressing video conferences to extremely low bitrates. Similarly, Gemino (Sivaraman et al., 2024) transmits video streams at extremely low resolution. At the receiver side, it estimates facial motion fields from the low-resolution video, then utilizes the motion fields and high-resolution facial images to generate high-resolution facial videos. Instead of driving static images, Reparo (Li et al., 2023) proposes a token-based generative streaming. It first uses VQGAN (Esser et al., 2021) to train a codebook of facial visual features, then maps the video to latent variables using VAE (Kingma & Welling, 2014), and finally quantizes the latent variables into tokens according to the codebook. Since tokens are indices of the codebook, the bitrate is extremely low. In general, generative streaming can greatly compress videos, but it is often designed for specific tasks (such as video conferencing) and lacks generality.

### 2.2 STABLE DIFFUSION

Stable Diffusion (Rombach et al., 2022; sd, 2024) is the most popular open-source text-to-image generative model. It learns a denoising process from 5.85 billion image-text pairs (LAION, 2024), enabling it to generate high-quality images by denoising the pure noise images. Different noise images lead to different generated images. Since the noise images are randomly sampled from Gaussian distribution, the generated images are random and uncontrollable. Thus, to control the content of the generated images, Stable Diffusion also receives user-input prompts (in natural language) as the condition for denoising, thereby generating images that align with the semantic descriptions of the prompts. Specifically, let $p$ represent the user-input prompt. Stable Diffusion first performs word embedding and semantic extraction using the CLIP model (Radford et al., 2021), converting discrete

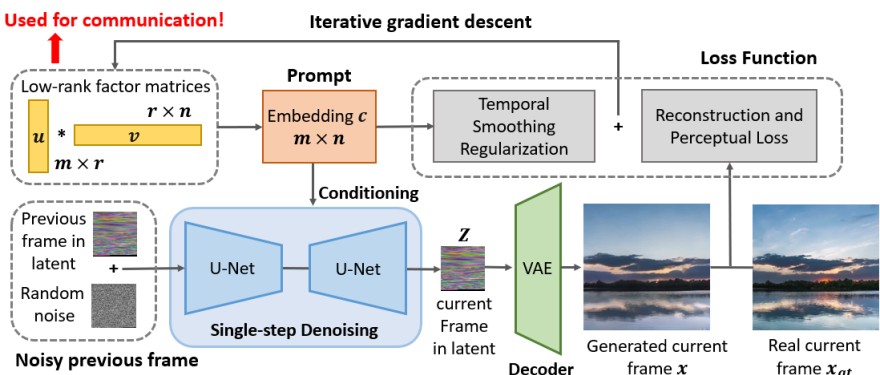

Figure 2: Workflow of Promptus's video to prompt inversion.

natural language $p$ into a continuous text embedding $c$ (an $m * n$ matrix). The text embeddings $c$ and the randomly sampled noise image $N$ are input into the denoising process of Stable Diffusion, generating the denoised $N'$. Next, $N'$ is input into the denoising process again for $T$ iterations to obtain the denoised $Z$. Since the denoising process occurs in the latent space, $Z$ needs to be input into the VAE Decoder (Kingma & Welling, 2014) to generate the image $x$ in the pixel space.

However, due to the inability to precisely control image generation, Stable Diffusion cannot meet the fidelity requirements of video streaming. Stable Diffusion only defines the generation process from prompt to image, without considering the inverse process of extracting prompts from images. The most straightforward solution is to extract text descriptions for target images using Image/Video Captioning algorithms (Yang et al., 2023; Hu et al., 2022) and use these extracted descriptions as prompts to generate images. Although the generated images can semantically align with the target images, these extracted descriptions are always too high-level, resulting in generated images having large structural differences, as shown in Figure 1. Besides text, ControlNet (Zhang et al., 2023) can also use images (such as masks) as prompts, controlling Stable Diffusion to generate images that align with the contours of the image prompts. However, apart from the contours, details such as color and texture cannot be aligned. In conclusion, the frames generated by Stable Diffusion cannot faithfully reproduce the original ones at the pixel level, making them unsuitable for video streaming.

Promptus is an initial effort to invert frames into prompts while ensuring pixel-level alignment. Promptus is related to some works on Text Inversion (Gal et al., 2022; Ruiz et al., 2023; Kawar et al., 2023), both of which learn prompts from images in an inverse manner. However, Text Inversion aims to learn text embeddings that represent the appearance of specific objects from multiple images, aligning only at the semantic level. Text Inversion has also been used for image compression (Pan et al., 2022). But similarly, the inverse prompt is only used for semantic alignment, while pixel alignment is achieved by using low-resolution images as conditions.

## 3 VIDEO TO PROMPT INVERSION

### 3.1 GRADIENT DESCENT BASED PROMPT FITTING

**Gradient Descent Framework.** To obtain the inverse prompt, the most straightforward approach is to train a neural network to map the target image to a prompt. However, on one hand, this approach makes it difficult to ensure pixel alignment (like the aforementioned Captioning algorithm (Yang et al., 2023; Hu et al., 2022)). On the other hand, as the inverse process of Stable Diffusion, this neural network needs to learn comparable knowledge, but this is very expensive (e.g., training a Stable Diffusion will cost between 600,000 and 10 million US dollars (SDc, 2024)). Therefore, instead of training a new neural network, we propose fully leveraging Stable Diffusion's knowledge to infer the prompt. To this end, we adopt gradient descent to iteratively fit the prompt, with the framework shown in Figure 2. Specifically, at the beginning, the prompt is randomly initialized. Stable Diffusion generates a frame based on this prompt. Since the prompt is random, the generated frame is meaningless. Third, the pixel-wise difference between the generated frame and the target frame is calculated as the loss value. Fourth, backpropagation is used to compute the gradient of the loss with respect to the prompt. Finally, the prompt is updated using gradient descent. The above steps are iteratively executed until the loss is sufficiently small, and the resulting prompt can satisfy

pixel-aligned generation. In the above steps, Stable Diffusion is pre-trained and frozen, so it has prior knowledge. This knowledge is gradually distilled into the prompt via gradient descent fitting.

To realize the aforementioned framework, there are several key components:

**Single-step denoising to avoid higher-order derivatives.** As described in §2.2, Stable Diffusion uses iterative denoising to generate images. Therefore, the prompt recursively affects the generated image. This causes the gradient of the loss value with respect to the prompt to involve the computation of higher-order derivatives (such as 20th order), which introduces prohibitive computational and memory overhead. Thus, instead of using the traditional Stable Diffusion, we adopt SD (2.1) Turbo (sdt, 2024), a variant that can generate frames through single-step denoising. The adoption of SD Turbo allows gradient descent to only compute the first-order derivatives, greatly improving efficiency. Although the quality of the generated frames is weaker than the traditional Stable Diffusion, these quality losses can be compensated for in an end-to-end manner during fitting.

**Employing embeddings as prompts to avoid non-differentiability.** Computing the gradient of the prompt through backpropagation requires the forward computation from the prompt to the loss value to be completely differentiable. However, as described in §2.2, the forward computation includes the CLIP module (Radford et al., 2021), which converts text from discrete natural language to continuous text embeddings. This conversion involves indexing and table lookup, making it non-differentiable. Gradients cannot be propagated to the text in natural language, preventing gradient descent. To address this, we discard the non-differentiable CLIP module and directly use text embeddings as prompts for conditioning Stable Diffusion. In this case, the forward computation is fully differentiable, allowing gradient descent to be performed on the text embeddings. In the following sections, prompt refers to the text in embedding rather than the text in natural language[1].

**Using a noisy previous frame instead of random noise to reduce latent space distance.** According to §2.2, in addition to the prompt, the input random noise also affects the generated frames. With the same prompt, different input noise results in different generated frames. At a high level, the input noise can be viewed as a point in the latent space, and the denoising process actually moves this point under the control of the prompt. Therefore, the goal of Promptus is to find the inverse prompt that can move the point represented by the noise to the point of the target image in the latent space. Since we adopt a single-step denoising Stable Diffusion, if the input noise is far from the target image in the latent space, this movement cannot be completed in a single step, making it impossible to fit the inverse prompt. As shown in Figure 3(a), using random noise as input, after the loss value converges, the generated frame still has noticeable differences from the target frame, including blurring, noise artifacts, and inconsistent details. Therefore, we need to reduce the distance between the input noise and the target image in the latent space. We observe that in a video, adjacent frames are close in the latent space. Thus, we manually add noise to the previous frame as follows:

$$N^t = (1 - \gamma) * Z^{t-1} + \gamma * N^0 \tag{1}$$

Where $Z^{t-1}$ is the previous frame in the latent space. $N^0$ is a fixed noise. $\gamma$ is a hyperparameter that controls the degree of noise addition, which we set to 0.95 in the experiment. $N^t$ is used as the noise input to Stable Diffusion for generating the current frame. As shown in Figure 3(c), compared to random noise, the noisy previous frame can reduce the distance in the latent space, resulting in a generated frame that better matches the target frame.

**Combining reconstruction and perceptual loss functions.** To achieve pixel-aligned supervision, the most intuitive loss function is the per-pixel reconstruction loss, such as MSE. These reconstruction losses attempt to minimize the error of each pixel, while errors in high-frequency details and edges often lead to large pixel errors. Therefore, to reduce the overall error, reconstruction losses tend to abandon the fitting of edges and details, resulting in overly smooth and blurry images, as shown in Figure 3(b). To make the generated frames sharp and clear, one approach is to use perceptual loss instead of reconstruction loss, such as LPIPS (Zhang et al., 2018). Perceptual loss is based on deep learning and can estimate the subjective quality of images as perceived by the human eye.

---

[1]It is important to note that, even though it is not natural language, embeddings can still serve as prompts. On one hand, prompts can take multiple modalities, such as images (Blattmann et al., 2023), audio (Biner et al., 2024), embeddings (Gal et al., 2022; Ruiz et al., 2023; Kawar et al., 2023), fMRI (Chen et al., 2024), etc. On the other hand, the fundamental characteristic of prompts is that they serve as conditions to guide generation. The resolution of the generated videos is only dependent on the size of the random noise, unrelated to the prompt size. So prompts have the advantage of being decoupled from video resolution compared to encoding.

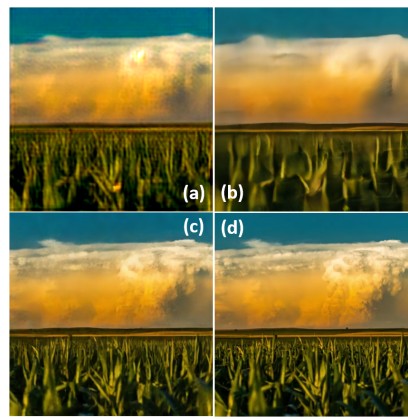

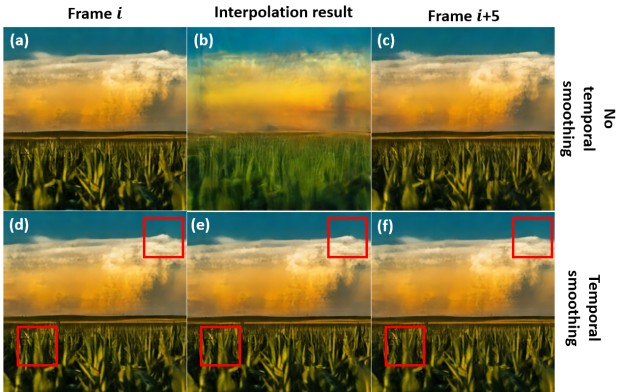

Figure 3: Visualization of prompt fitting results. (a) Using random noise as input. (b) Only using MSE as the loss function. (c) Ours. (d) Ground Truth.

Figure 4: Visualization of the prompt interpolation results. When applying temporal smoothing regularization, the interpolation results not only fully preserve the video details but also successfully approximate the motion in the video.

Since the human eye is highly sensitive to image details, perceptual loss can make the generated images sharper with richer details. However, perceptual loss aims to maximize the overall subjective quality of the image without focusing on the exact consistency of each pixel, leading to misalignment between the generated and target images. Thus, to simultaneously ensure pixel alignment and subjective quality, we combine the reconstruction and perceptual loss as the fitting loss $D$:

$$D = \alpha * D_{rec}(x, x_{gt}) + (1 - \alpha) * D_{per}(x, x_{gt}) \tag{2}$$

where $D_{rec}$ represents the reconstruction loss, which is MSE by default. $D_{per}$ represents the perceptual loss, which is LPIPS by default. $\alpha$ is a hyperparameter that balances pixel alignment and perceptual quality. In our experiments, we set $\alpha$ to 0.8. The final result is shown in Figure 3(c). It can be observed that by jointly optimizing the reconstruction loss and the perceptual loss, the image generated by Promptus ensures pixel-level alignment while maintaining a sharp appearance.

## 3.2 Low-rank Decomposition based Prompt Bitrate Control

According to §3.1, each prompt is an $m*n$ embedding matrix (e.g., $1024*77$), where each element is a floating-point number (e.g., 32-bit float), resulting in a fixed and high bitrate. To precisely control the bitrate of prompts, Promptus has two directions: First, dimensionality reduction decreases the number of prompt parameters. Second, quantization reduces the number of bits for each parameter.

**Low-rank matrix decomposition.** To perform dimensionality reduction on the prompt, the most straightforward method is to first fit the complete prompt and then perform dimensionality reduction algorithms such as Singular Value Decomposition or Principal Component Analysis on it. However, these methods only perform dimensionality reduction based on the data distribution of the prompt, ignoring the impact of prompt degradation on the generation results. This inevitably leads to a degradation in the quality of the generated images. Therefore, instead of performing explicit dimensionality reduction, Promptus proposes to directly fit a low-dimensional prompt end-to-end, thereby reducing the quality degradation caused by dimensionality reduction. To achieve this, Promptus integrates the inverse process of CANDECOMP/PARAFAC decomposition (Harshman et al., 1970) into gradient descent fitting. Specifically, Promptus calculates the embedding $c$ as follows:

$$c = \frac{u * v}{\sqrt{r}} \tag{3}$$

where $u$ and $v$ are two low-rank factor matrices with dimensions $m*r$ and $r*n$, respectively. $r$ is the rank of the embedding. $u$ and $v$ compose the embedding $c$ through outer product and normalization. At this point, the embedding $c$, as an intermediate variable, is no longer fitted or stored. $u$ and $v$, as the new representation of the prompt, will be randomly initialized and fitted.

The rank $r$ determines the trade-off between bitrate and quality. Compared to the embedding size of $m * n$ (e.g., $1024 * 77$), the total size of $u$ and $v$ is $(m + n) * r$. Therefore, reducing $r$ significantly

lowers the bitrate. However, on the other hand, when Rank $r$ is smaller, the embedding is constrained to be a low-rank matrix, resulting in weaker representational capability and inability to fit high-frequency details in the image, as shown in Figure 5. So it is necessary to dynamically select the most appropriate $r$ based on the current network bandwidth to trade off between bitrate and quality.

**Fitting-aware quantization.** Although the number of parameters in the prompt has been significantly reduced through low-rank matrix decomposition, each parameter in $u$ and $v$ is still a high-bit floating-point number (such as 32-bit float type). Therefore, to further reduce the bitrate, it is necessary to quantize $u$ and $v$, reducing the number of bits for each parameter. We adopt differentiable fake quantization (Zafrir et al., 2019) and incorporate quantization into the fitting process, automatically compensating for the quantization loss through end-to-end gradient descent. We test the impact of different quantization configurations on quality. The results indicate our method can reduce the number of bits from 32 to 8 with almost no quality loss.

### 3.3 Prompt inter-frame compression based on temporal smoothing

According to §3.1, Promptus fits each frame of the video as an independent prompt, without considering the correlation across frames. Therefore, during video streaming, Promptus needs to transmit a prompt for each frame, resulting in a linear increase in bitrate as the frame rate rises. However, codec-based streaming can avoid this problem through inter-frame compression. Is it possible to perform inter-frame compression on prompts as well? The most straightforward solution is to reshape the prompt of each frame into a two-dimensional matrix and encode it using a video codec. However, unlike conventional videos, we found that this reshaped prompt looks like random noise, lacking any patterns, structures, or smooth regions, causing video codecs to no longer work.

For inter-frame compression of prompts, our insight is: prompts are high-level semantics, so the prompts of continuous video frames should change continuously. If two temporally close frames are also sufficiently close in the prompt space, then the prompts of the frames between these two frames can be approximated by linear interpolation. With this, during streaming, we only need to sparsely transmit the prompts of a few frames (as keyframes), and the prompts of the remaining frames can be obtained by linear interpolation of the keyframe prompts. Specifically, we defines the keyframe interval as $K$ (the choice of $K$ is discussed in §4.2), adding one keyframe every $K$ frames. Since keyframes account for only a small portion of the total frames, inter-frame compression is achieved.

To ensure that adjacent frames are sufficiently close in the prompt space, we add temporal smoothing regularization to the embedding during fitting, as follows:

$$\lambda = \left\| c^t - c^{t-1} \right\|_2 \tag{4}$$

Here, $c^t$ represents the embedding of the frame currently being fitted, while $c^{t-1}$ denotes the embedding of the previously fitted frame. The final loss function $L$ is as follows:

$$L = \beta * D + (1 - \beta) * \lambda \tag{5}$$

Where $\beta$ is a hyperparameter used to balance the fitting loss and the temporal smoothing regularization. In our experiments, we set it to 0.2.

Temporal smoothing regularization works, as shown in Figure 4. Without temporal smoothing regularization, the interpolation results suffer from severe distortions, disrupting the video's content and structure. With temporal smoothing regularization, the interpolation results not only fully preserve the video details but also successfully approximate the motion in the video.

In practice, scene changes often occur within the video, at which point interpolation no longer works. Therefore, we will continuously detect scene changes and treat the new scenes as new videos (§A.1). For more details on the system design and implementation, please refer to §A if interested.

## 4 Evaluation

### 4.1 Experiment Setup

**Test videos**: To validate the generalizability across different domains, we selected 7 videos from 4 datasets with vastly different content, as summarized in Table 1. The domains of these videos span natural landscapes and human activities, outdoor long-range scenes and indoor close-up scenes, real-world scenes and CG-synthesized scenes, 3D video games and 2D animations. All videos are cropped and resized to a resolution of 512*512, with a frame rate of 30 FPS. Note that the resolution does not have to be 512*512, as Stable Diffusion supports flexible resolution for generation.

Table 1: Time overhead of generating a 512*512 frame. Detail in §C.

| Steps | Time (ms) |
|---|---|
| Dequantization | 0.016 |
| Prompt composition | 0.025 |
| Prompt interpolation | 0.013 |
| Noised frame | 0.012 |
| Stable Diffusion generation | 6.160 |
| Total | 6.226 |

Table 2: Test videos summary

| Dataset | #videos, frames | Description |
|---|---|---|
| QST Zhang et al. (2020) | 2, 300 | Natural landscapes, outdoor distant view |
| UVG Mercat et al. (2020) | 2, 300 | Human activity, face, hand, indoor close-up |
| GTA-IM Cao et al. (2020) | 2, 300 | 3D Game recording, CG-synthesized |
| Animerun Siyao et al. (2022) | 1, 60 | 2D animation, cartoon |
| Total | 7, 960 | |

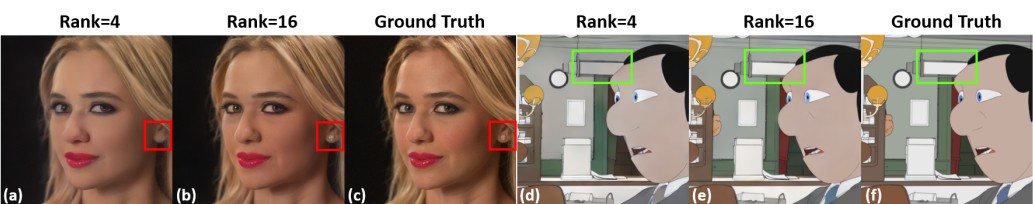

Figure 5: Visualization of the fitting results at different ranks. It can be seen that as the rank increases, the prompt can fit more details. For example, when the rank is 4, the earrings in (a) are lost, and the lamp's color in (d) is inconsistent with the ground truth. When the rank increases to 16, the earrings in (b) are successfully fitted, and the lamp's color in (e) is also corrected.

**Baselines**: We compare Promptus with three baselines: H.265 (265, 2024), H.266 (Wieckowski et al., 2021) and VAE (Rombach et al., 2022; Kingma & Welling, 2014). H.265 and H.266 are advanced traditional codecs that achieve compression by utilizing hand-designed modules. VAE is a deep learning-based neural codec. Both its encoder and decoder are trainable neural networks. The encoder maps the input image to low-dimensional latent variables, while the decoder reconstructs the image from the latent variables. The dimensionality of the latent variables is usually much smaller than that of the original image, thus achieving compression. For inter-frame compression of VAE, we encode the latent variables into a video using H.265. To balance the quality and bitrate of VAE, we adjust the target bitrate of video encoding. The training set of VAE is the same as that of SD.

**Metric**: To evaluate video quality, we adopt the LPIPS (Zhang et al., 2018) instead of the traditional SSIM and PSNR. This is because LPIPS better reflects human subjective perception of video quality (Yu et al., 2023; Zhang et al., 2018). A smaller LPIPS value indicates a higher quality.

### 4.2 TRADE OFF BETWEEN BITRATE AND QUALITY

**Prompt rank.** We illustrate the variation in visual quality of Promptus under different ranks, as depicted in Figure 6. It indicates that the higher the prompt rank, the higher the quality of Promptus. For example, when the prompt rank increases from 4 to 16, the LPIPS decreases from 0.265 to 0.221 (on the line with a keyframe interval of 1). This is because the larger the rank, the stronger the representational capability of the prompt, allowing it to fit the target image more accurately, as described in §3.2. We present a visualization of the fitting results at different ranks in Figure 5. It can be seen that as the rank increases, the prompt can fit more details. For instance, when the rank is 4, the earrings are lost, while when the rank increases to 16, the earrings are successfully fitted.

**Keyframe interval.** Figure 6 also shows the impact of different keyframe intervals on quality. The smaller the keyframe interval, the higher the quality of Promptus. For example, when the prompt rank is 16, reducing the keyframe interval from 10 to 1 decreases the LPIPS from 0.274 to 0.221. This is because, when the keyframe interval is smaller, the distance between keyframes in the prompt space is smaller. At this time, the linear interpolation of keyframe prompts can more accurately approximate the prompts of intermediate frames, as described in §3.3.

**Quality-bitrate tradeoff.** Increasing the prompt rank and reducing the keyframe interval lead to a rapid rise in bitrate while improving quality. Therefore, we present the tradeoff between bitrate and quality, as shown in Figure 7. First, it illustrates that overall, the quality monotonically increases

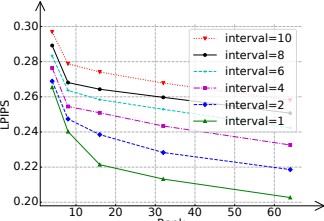 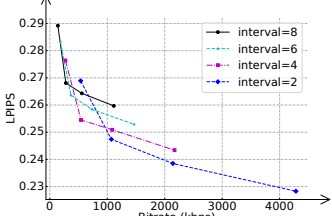 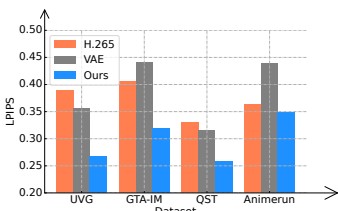

Figure 6: The impact of prompt rank and keyframe interval on visual quality. It indicates that visual quality improves with increasing prompt rank and decreasing keyframe interval.

Figure 7: The tradeoff between bitrate and quality for Promptus. The quality monotonically increases with the increase in bitrate. Besides, Promptus can achieve scalable bitrates.

Figure 8: Average frame quality on 4 datasets. It proves the generality of Promptus across domains. The more high-fidelity details a video has, the greater the advantage of Promptus.

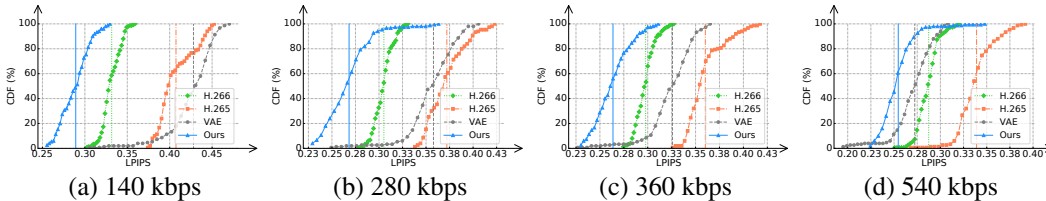

| (a) 140 kbps | (b) 280 kbps | (c) 360 kbps | (d) 540 kbps |

Figure 9: Frame quality CDF and mean of Promptus and baselines at four target bitrates. It indicates Promptus achieves better compression efficiency across all bitrate levels. Detail in §4.3.

with the increase in bitrate. Thus, to optimize quality, Promptus sends prompts at a bitrate closest to the available bandwidth. Second, Promptus can achieve scalable bitrates. For example, by adjusting the rank in the range of 4 to 32 and the keyframe interval in the range of 2 to 8, Promptus's bitrate spans from 113 kbps to 4284 kbps. Third, at the same bitrate, the quality varies for different configurations. For instance, when the bitrate is 550 kbps, the parameter configuration with a rank of 8 and a keyframe interval of 4 has an LPIPS of 0.255, while the parameter configuration with a rank of 16 and a keyframe interval of 8 has an LPIPS of 0.264, which is lower in quality than the former. So at the same bitrate, we tend to choose parameter configurations with smaller keyframe intervals.

## 4.3 COMPRESSION EFFICIENCY

This section demonstrates the compression efficiency. Figure 9 shows the CDF of the frame quality under 4 bitrate levels. Since a lower LPIPS represents better visual quality, a leftward shift of the curve in the figure represents more high-quality frames, and thus higher compression efficiency. We also calculate the average LPIPS for each method, represented by the vertical lines in the figure.

First, Promptus achieves better compression efficiency across all bitrate levels. For example, in Figure 9, the curves of Promptus are all to the left of the baseline curves. Second, Promptus can achieve more than a 4x bandwidth reduction while preserving the same perceptual quality. For example, the average LPIPS of Promptus under 140 kbps is better than H.265 under 540 kbps. Third, the lower the bitrate, the greater the advantage of Promptus. At a high level, as the bitrate decreases from 540 kbps to 140 kbps, the distance between Promptus's curves and the baselines' curves gradually widens. At a low level, when the bitrate is 540 kbps, the average LPIPS of Promptus is 0.018, 0.085 and 0.033 lower than VAE, H.265 and H.266, respectively. When the bitrate is 140 kbps, this difference further increases to 0.139, 0.118 and 0.042. This is because when the bitrate is reduced, the traditional codecs use coarser quantization and lose many high-frequency details, resulting in blurriness and block artifacts in the video, which significantly impairs the perceptual quality. VAE mitigates this phenomenon by mapping images to a low-dimensional latent space, reserving more bitrate for video encoding. Therefore, at most bitrates, the quality of VAE is superior to H.265. However, at extremely low bitrates (such as 140 kbps), the latent space inevitably introduces distortions caused by video coding. At this point, the VAE Decoder introduces a large number of errors when reconstructing the images, causing a significant decrease in VAE's quality. On the other hand, **when the bitrate is reduced, Promptus reduces the representational capability of the prompt rather than degrading the video quality**. This prevents Promptus from accurately describing the video content, resulting in slight misalignments in the gen-

erated frames. However, thanks to the inherent image generation capability of Stable Diffusion, Promptus's frames still have good sharpness and details, thus outperforming the three baselines.

### 4.4 GENERALITY

This section proves the generality of Promptus across different domains. Figure 8 shows the average frame quality for Promptus and two baselines on four datasets. First, Promptus has better compression efficiency on different domains. As shown in Figure 8, Promptus achieves lower average LPIPS compared to the baselines on each dataset. To intuitively demonstrate this gain, we visualize the compression results of each method on the four datasets at low bitrates (such as 225 kbps) in Figure 10. It can be observed that, compared to the baselines which exhibit blurriness and blocking artifacts, Promptus preserves more high-frequency details, resulting in higher perceptual quality.

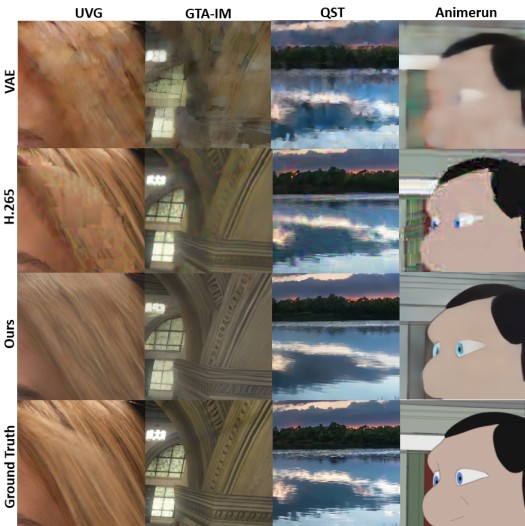

Figure 10: Visualization of the compression results on different datasets. It can be observed that, compared to the baselines which exhibit blurriness and blocking artifacts caused by compression, Promptus preserves more high-frequency details.

Second, the more high-frequency details a video has, the greater the advantage of Promptus. For example, for the Animerun dataset with fewer details, the LPIPS of Promptus is 0.015 lower than H.265, which is not a significant advantage. However, for the detail-rich UVG, this difference further expands to 0.121. This is because for 2D animations with large areas of solid colors and simple details, H.265's intra-frame prediction, block partitioning, and motion compensation techniques can handle them well, so Promptus's performance gain is small. For detail-rich real-world videos, H.265 discards more high-frequency information during compression, thus damaging the perceptual quality. Although Promptus also loses high-frequency information, Stable Diffusion completes some lost information based on prior knowledge during generation, resulting in a smaller quality loss.

We conduct evaluations under real-world network conditions (§B), and the results demonstrate Promptus can decrease the ratio of severely distorted frames (whose LPIPS is higher than 0.32) by 89.3% and 91.7% comared to VAE and H.265. We perform overhead evaluations (§C), and the results show that Promptus can generate video in real-time. We conduct ablation studies on interpolation methods and the generator (§D), and the results demonstrate the U-Net (diffusion process) plays a significant role in compression performance. We present the X-t slices of the videos (Figure 11), and the results show that our videos are aligned with the ground truth videos in terms of motion. We show the fitting results for some challenging examples (Figure 16), demonstrating that Promptus can successfully fit elements that are difficult for Stable Diffusion itself to generate.

## 5 DISCUSSION

In this paper, we propose Promptus, a novel system that replaces video streaming with prompt streaming by representing video frames as Stable Diffusion prompts. To ensure pixel alignment, a gradient descent-based prompt fitting framework is proposed. To achieve adaptive bitrate, a low-rank decomposition-based bitrate control algorithm is introduced. For inter-frame compression, a temporal smoothing-based prompt interpolation algorithm is proposed. Evaluations across various video domains and real network traces demonstrate that, compared to H.265, Promptus can achieve more than a 4x bandwidth reduction while preserving the same perceptual quality.

Promptus extends the boundaries of AIGC to video streaming, offering a new communication paradigm. As an initial effort, the current version has some limitations, such as the time overhead of prompt fitting and the latency of prompt interpolation (details are discussed in §E). These limitations restrict Promptus to on-demand videos, preventing its use in live videos. Therefore, further improving the fitting efficiency and reducing the prompt bitrate are future works.

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

## A    PROMPT STREAMING SYSTEM DESIGN AND IMPLEMENTATION

§3 shows how Promptus inverts videos into pixel-aligned prompts. In this section, we will further describe how Promptus utilizes this Prompt Inversion technique to design a streaming system. Specifically, it will be divided into two parts: the sender side and the receiver side.

### A.1    THE SENDER SIDE

On the sender side, Prompt Inversion replaces video encoding. At a high level, Promptus performs Prompt Inversion on raw frames as target images. The obtained prompts are then streamed to the receiver side for image generation and playback. The details are as follows:

**Initialization for the first frame.** As stated in §3.1, to improve the quality of the generated images, Promptus uses the noisy previous frame as the input noise for Stable Diffusion. However, for the first frame of the video (the frame index starts from 1), there is no previous frame $Z^0$. To address this, Promptus uses the VAE's Encoder to map the first frame itself to the latent space, obtaining $Z^0$. Then, noise is added to $Z^0$ according to Equation 1. The noisy $N^1$ will be used as the input noise for Prompt Inversion of the first frame. Since the purpose of using the noisy previous frame is to reduce the distance between the input noise and the target image in the latent space, the noisy current frame naturally works as well. Note that this noise $Z^0$ will also be sent to the receiver side along with the prompt for generating the first frame image. On the other hand, due to the absence of a previous frame, Promptus does not apply temporal smoothing regularization for the Fitting of the first frame.

**Re-initialization for abrupt scene changes.** When the video undergoes drastic changes, such as suddenly switching to a new scene, the content difference between the previous frame and the current frame becomes significant, and the distance in the latent space is no longer close. In this case, using the noisy previous frame as described in §3.1 will no longer work. Therefore, Promptus detects abrupt scene changes. Specifically, Promptus calculates the distance between the current frame and the previous frame in the latent space, and when this distance exceeds a threshold, it is considered an abrupt scene change. In this situation, Promptus treats the video after the scene change as a new video and performs the aforementioned first frame initialization again.

**Sparse prompt streaming.** As stated in §3.3, through temporal smoothing regularization, the prompts of most frames can be approximated by linear interpolation of the prompts of keyframes.

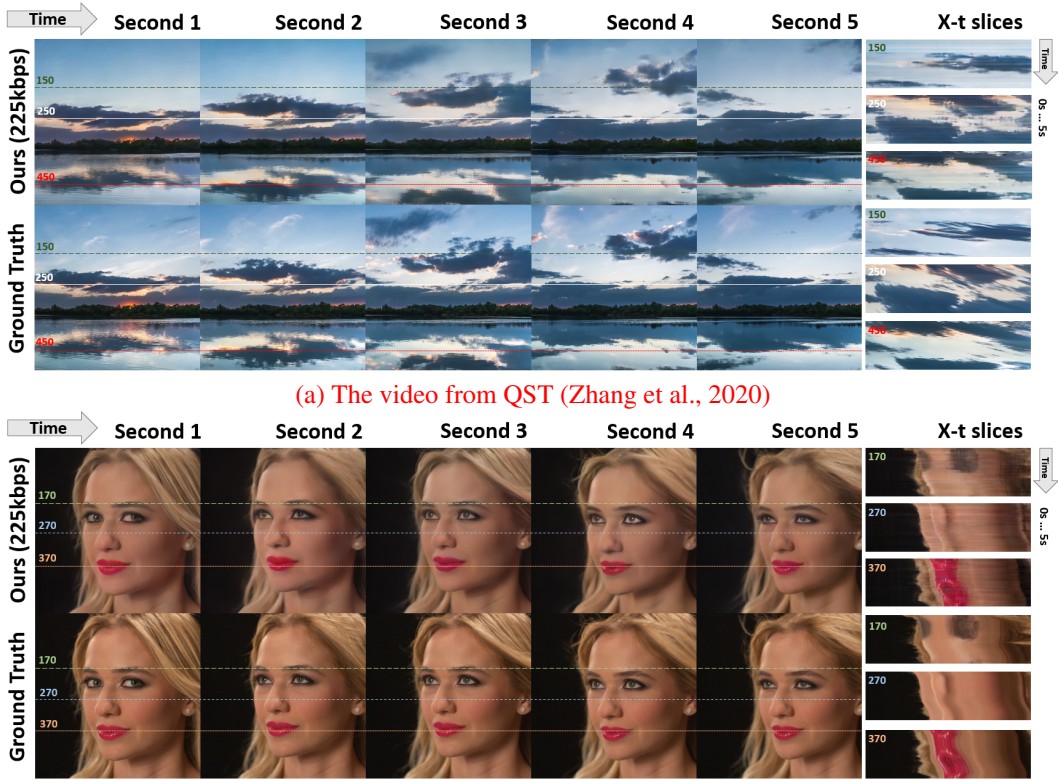

(a) The video from QST (Zhang et al., 2020)

(b) The video from UVG (Mercat et al., 2020)

Figure 11: To evaluate the temporal consistency of the generated videos, we show the videos second by second. We also visualize the entire videos as space-time X-t slices (Li et al., 2024b), with three scanning lines at different heights marked on the left. Both of our videos are inversed at a bitrate of 225 kbps. The results indicate that our videos are temporally aligned with the ground truth videos in terms of motion.

Thus, Promptus adds one keyframe every $K$ frames. Only the prompts of keyframes will be streamed to the receiver. Note that when the aforementioned abrupt scene changes occur, the difference between frames in the latent space is significant, and the interpolation of prompts no longer works. In this case, the last frame before the scene change will also become a keyframe. The frames after the scene change will start a new count.

**Low-rank based adaptive bitrate.** According to §3.2, the higher the bitrate of the prompt, the higher the quality of the generated image. Therefore, in prompt streaming, it is necessary to increase the prompt bitrate as much as possible while avoiding network congestion to maximize the user experience. Consequently, Promptus fits multiple prompts of different ranks in advance. During streaming, Promptus selects the prompt with the bitrate closest to the estimated bandwidth for transmission. It is worth noting that the prompts transmitted by Promptus are not encoded. Therefore, compared to codec-based streaming, Promptus can precisely control the bitrate.

## A.2 THE RECEIVER SIDE

On the receiver side, Stable Diffusion's prompt-to-image generation replaces video decoding. The details are as follows:

**Video generation based on sparse prompts.** As shown in Figure 12, after receiving the prompt of keyframe $i$, the receiver first performs linear interpolation between the prompts of adjacent keyframes $i - k$ and $i$ to approximate the prompts of the intermediate $K - 1$ frames. Afterward, these prompts are used for Stable Diffusion's image generation. As described in §3.1, the generated $Z$ for each frame will be noised and used as the input noise for the generation of the next frame.

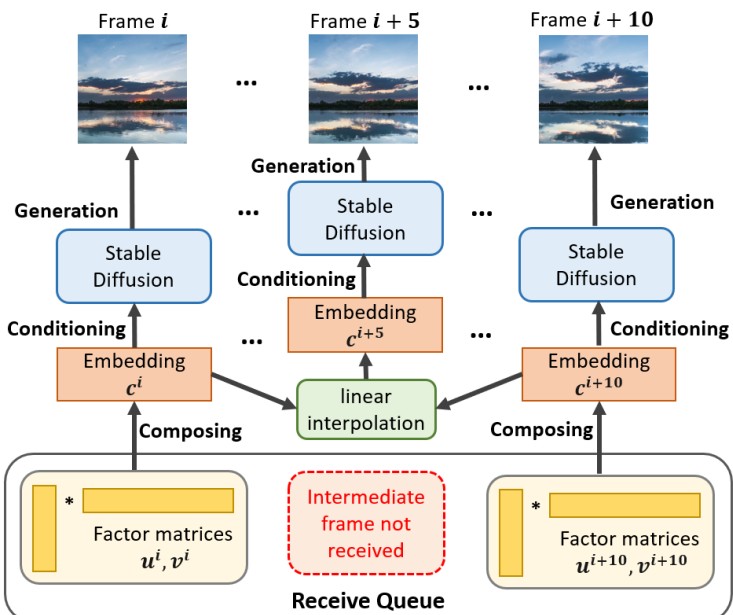

Figure 12: At the receiver, Promptus approximates the unreceived frames by linear interpolation in the prompt space. Thus, instead of sending the prompt for each frame, only the prompts for key frames need to be transmitted sparsely, thereby achieving inter-frame compression.

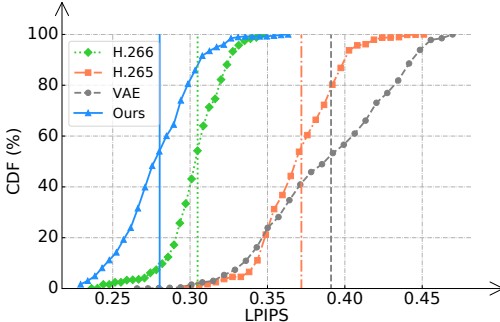

Figure 13: The performance of Promptus under real network traces. On one hand, Promptus's quality is overall higher than the baselines. On the other hand, Promptus can significantly reduce the ratio of severely distorted frames (whose LPIPS is higher than 0.32).

**Real-time video generation.** The frame rate of video playback at the receiver depends on Promptus's image generation speed, making real-time generation crucial. Promptus's image generation consists of five parts: prompt dequantization, prompt composition, prompt interpolation, adding noise to previous frame and Stable Diffusion image generation. The first four parts only involve simple linear calculations, so their time consumption can be ignored. The speed of Stable Diffusion becomes the bottleneck. To address this, we follow StreamDiffusion (Kodaira et al., 2023), using TAESD and TensorRT to accelerate the Stable Diffusion generation. By this, Promptus achieves real-time video generation at the receiver.

## B  PERFORMANCE ON REAL-WORLD TRACES

This section demonstrates the performance of Promptus under real network traces. We collected 6 network traces from real-world scenarios such as subways, driving, and walking, under 2G, 3G, and 4G networks. The traces consisted of 1 from a 4G network, 2 from 3G, and 3 from 2G. Each

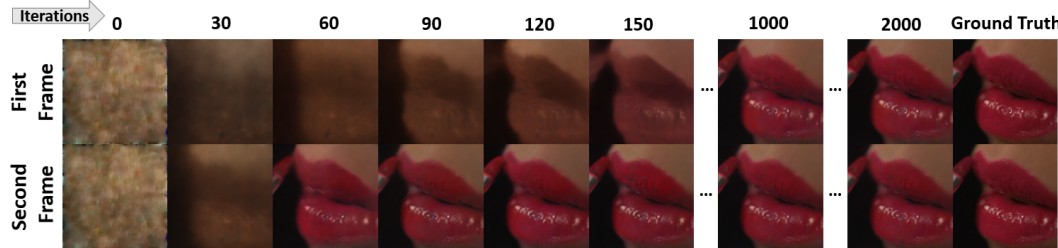

Figure 14: Visualization of fitting results at different iterations. The results indicate that for the first frame of a new video (or new scene), the fitting process takes thousands of iterations to converge since it starts from scratch. For subsequent frames, thanks to the temporal smoothing regularization §3.3 providing sufficient scene priors, the fitting process can converge in only a few hundred iterations. For example, the second frame in the figure requires only 120 iterations to achieve a visual quality comparable to that of 2000 iterations for the first frame.

trace lasted between 5 and 30 seconds, totaling over 100 seconds. Our traces aimed to test various weak network conditions such as poor signal coverage, network overload, high-speed movement, and frequent switching of mobile communication networks. The average bandwidth per second ranged between 50 kbps and 4000 kbps, while the one-way network delay was approximately 30 ms to 100 ms. We use Mahimahi (Netravali et al., 2015) to replay the these network traces. The queue length of Mahimahi is set to 60, and the drop-tail strategy is adopted. Since the total length of the traces is greater than the total length of the test videos, we loop the test videos to run through the entire traces.

Figure 13 shows the CDF and mean of the frame quality. First, Promptus's quality is overall higher than the baselines. For example, the mean LPIPS of Promptus is 0.111, 0.092 and 0.025 lower than VAE, H.265 and H.266, respectively. Second, Promptus can significantly reduce the ratio of severely distorted frames. For instance, only 5.2% of frames in Promptus have LPIPS higher than 0.32, while VAE and H.265 have 94.5% and 96.9%, respectively. These improvements are partly due to Promptus's excellent compression efficiency, enabling it to provide higher perceptual quality at the same bitrate. On the other hand, since Promptus sends raw prompts without encoding (such as entropy coding), it can precisely control the target, thus making full use of bandwidth.

## C OVERHEAD

In this section, we analyze the overhead of Promptus. We conduct tests on an Nvidia 4090D GPU, using CUDA to accelerate Promptus. The resolution of the generated video is 512*512, and the rank of the prompt is 8. The Stable Diffusion model has a total of 867M parameters. With more parameters (such as SDXL Turbo (sdx, 2024)), the Stable Diffusion model has stronger generative ability, making prompt fitting easier and allowing for higher compression rates. However, this also leads to increased overhead in terms of memory usage and run time. Thus, the current version of Promptus employs SD 2.1 Turbo, which is relatively lightweight.

**Generation overhead.** Table 1 shows the fine-grained overhead of each step in Promptus's image generation. Specifically, from receiving a prompt to generating a frame, Promptus includes the following steps: prompt dequantization, prompt composition, prompt interpolation, adding noise to the previous frame, and Stable Diffusion generation. Among them, most steps only involve simple linear computations, so the time overhead is almost negligible. In contrast, Stable Diffusion generation accounts for the vast majority of the time overhead. Therefore, following StreamDiffusion (Kodaira et al., 2023), we use TAESD and TensorRT to accelerate the Stable Diffusion to 6.16 ms. In summary, the total time overhead of Promptus's image generation at the receiver is 6.226 ms, achieving real-time. Besides, the total memory usage is 8952 MB.

**Inversion overhead.** The time overhead of inversion depends on two factors: the number of iterations needed for convergence and the time taken for each iteration. Among them, the time taken for each iteration is 150 ms. For the number of iterations, there is a significant difference between different frames. For the first frame of a new video (or new scene), the inversion takes thousands

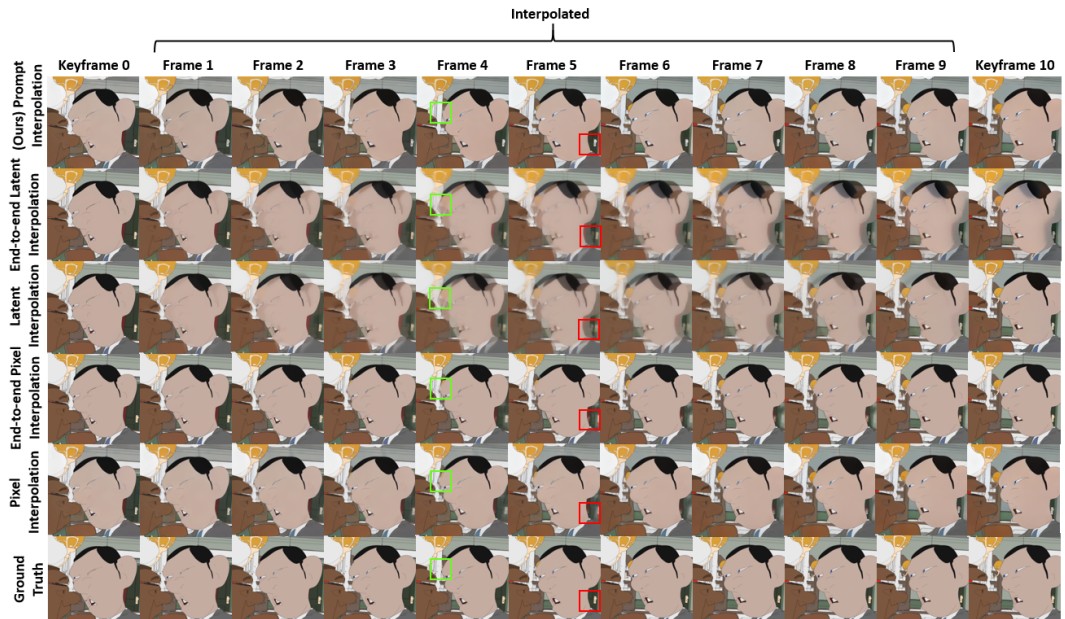

Figure 15: Comparison of different interpolation methods. Each method is based on frame 0 and frame 10 to interpolate frames 1 to 9. The results demonstrate that latent interpolation fails to preserve the motion between frames, resulting in spatial overlaps and ghosting. This is because the frames in latent space are not temporally close, making the interpolation unreasonable. While pixel interpolation (Huang et al., 2022) can preserve motion, it inevitably leads to noticeable artifacts in cases of occlusion, edges, or newly appearing objects due to incorrect matching, as illustrated by the green and red boxes in the figure. Our prompt interpolation successfully preserves motion while avoiding artifacts and keeping the edges sharp. Details can be found in §D

of iterations to converge since it starts from scratch. For subsequent frames, thanks to the temporal smoothing regularization §3.3 providing sufficient scene priors, the inversion can converge in only a few hundred iterations. We present an example in Figure 14. It can be seen that the second frame requires only 120 iterations to achieve a visual quality comparable to that of 2000 iterations for the first frame. In practice, we set the number of iterations for the first frame to 10,000, and for subsequent frames to 500. As for memory usage, inversion occupies a total of 18 GB.

# D    ABLATION STUDY OF INTERPOLATION METHODS AND GENERATORS

In this section, we compare different interpolation methods and analyze why it should be prompt interpolation rather than latent or pixel interpolation. We also demonstrate why it is necessary to use the complete Stable Diffusion as the generator for inversion rather than just the VAE (Decoder). Specifically, for latent interpolation, we remove the U-Net from the inversion framework (described in §3), retaining only the VAE Decoder. Thus, we randomly initialize the frames in the latent space and apply gradient descent with a temporal smoothing regularization for fitting. After finishing the fitting, we perform linear interpolation on the fitting results (latent variables) of the key frames to approximate the intermediate frames, just like in prompt interpolation. The results are shown in "End-to-end latent interpolation" in Figure 15. For pixel interpolation, we apply RIFE (Huang et al., 2022) (a real-time video frame interpolation algorithm) to the ground truth images of the key frames, directly estimating the intermediate frames. The results are shown in "End-to-end pixel interpolation" in Figure 15. Additionally, to compare with Promptus based solely on interpolation methods, we also conduct experiments where key frames (both the latent variables and images) come from Promptus, represented as "Latent interpolation" and "Pixel interpolation" shown in Figure 15. Each of the above methods is based on frame 0 and frame 10 to interpolate frames 1 to 9.

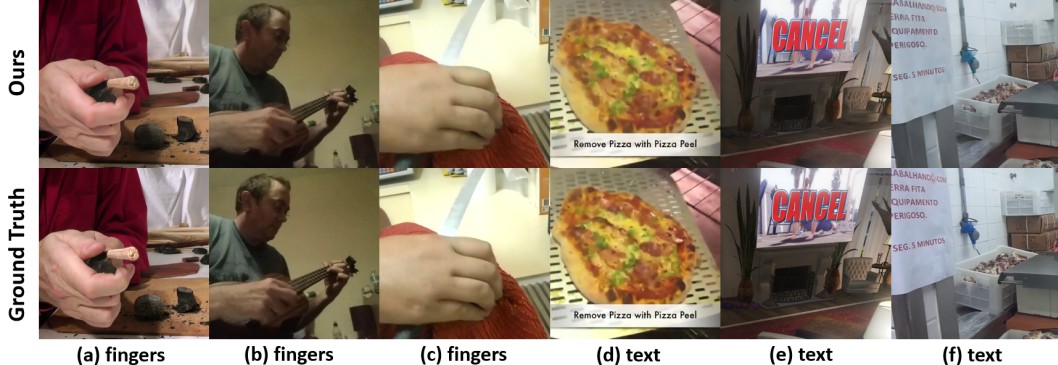

Figure 16: Fitting results for challenging examples, with a bitrate of 225 kbps. Although the Stable Diffusion itself struggles to generate specific elements, such as fingers and text, Promptus is able to fit them well. This is because end-to-end gradient descent fitting can compensate for the limitations of the Stable Diffusion itself.

First, the results demonstrate that latent interpolation fails to preserve the motion between frames, resulting in spatial overlaps and ghosting, as shown in the second and third rows of the Figure 15. This is because the frames in latent space are not temporally close, making the interpolation between frames unreasonable. This indicates that the VAE alone cannot serve as the generator for video inversion, since inter-frame compression does not work, and frame-by-frame transmission would result in significant bandwidth requirements. One feasible solution is to use a codec to encode the latent variables of each frame. However, as shown in Figure 9 and Figure 8, this solution performs worse than prompt interpolation. Moreover, the keyframe 10 of "End-to-end latent interpolation" looks worse than that of "latent interpolation". It again demonstrates that the frames are not temporally close in the latent space, enforcing a temporal smoothing regularization leads to artifacts in the fitting results.

Second, while pixel interpolation can preserve motion, it inevitably leads to noticeable artifacts in cases of occlusion, edges, or newly appearing objects due to incorrect matching, as illustrated by the green and red boxes in the Figure 15.

Third, our prompt interpolation successfully preserves motion while avoiding artifacts and keeping the edges sharp. This is because the U-Net converts frames from latent space to prompt space, where the frames are temporally close, allowing the interpolation to approximate motion. It proves that the U-Net (diffusion process) plays a significant role in compression performance.

## E   LIMITATION

**The time overhead of prompt fitting.** Although Promptus can achieve real-time video generation at the receiver, prompt fitting cannot be performed in real-time at the sender. This is because prompt fitting requires iterative gradient descent. Although a single iteration is fast, the total time overhead is high due to the large number of iterations required for convergence (such as 500 iterations). As a result, Promptus can currently only be used for Video on Demand and is not applicable for Live Video or Real-Time Communication. To address this, using more efficient gradient descent algorithms to reduce the number of iterations required for convergence is expected to accelerate prompt fitting to real-time.

**The latency of prompt interpolation.** At the receiver side, Promptus obtains the prompts of intermediate frames through prompt interpolation of keyframes. This means that if an intermediate frame needs to be generated and played, it is necessary to wait until the subsequent keyframe is received, which introduces additional latency. Although this latency has little impact on Video on Demand, as DASH (Sodagar, 2011) provides ample buffering to cover the interpolation latency. However, it is not suitable for latency-sensitive applications, such as WebRTC-based (raz, 2022) video conferencing and cloud gaming, where buffer sizes are small. To address this issue, designing keyframe extrapolation algorithms to replace interpolation is a future research direction.

**Non-uniform keyframes.** According to §A.1, Promptus sends keyframes uniformly based on the keyframe interval. However, different segments of a video often have different rates of change. For rapidly changing segments, the distance between keyframes in the prompt space is large, resulting in a poor approximation of intermediate frames using linear interpolation. A solution is to reduce the keyframe interval and send keyframes more densely. However, there also exist smoothly changing segments in the video, where densely sending keyframes brings little improvement to quality, thus wasting bandwidth. In summary, sending keyframes uniformly in this paper is inefficient. In the future, adaptive keyframe needs to be designed to transmit densely in rapidly changing segments and sparsely in smoothly changing segments.

