# OpenReview forum: "Promptus: Representing Real-World Video as Stable Diffusion Prompts for Video Streaming"
_ICLR.cc/2025/Conference — ICLR 2025 Conference Withdrawn Submission_

### Official Review · Reviewer_W5gp · 2024-10-24

**Soundness:** 3
**Presentation:** 3
**Contribution:** 3
**Rating:** 6
**Confidence:** 4

**Summary:**

This paper propose Promptus, a novel system that replaces video streaming with prompt streaming by representing video frames as Stable Diffusion prompts. To achieve this goal, this paper conducted experiments in three aspects: ensuring pixel alignment, achieving adaptive bitrate, and inter-frame compression. Experiments shown that Promptus can achieve more than 4x bandwidth reduction while perserving the same perceptual quality compared to H.265.

**Strengths:**

1.	Promptus provides a new communication paradigm, using only prompts to stream a video.
2.	The paper is easy to understand well and clearly written.
3.	The effect at low bitrates is significantly improved. Compared with VAE and H.265, Promptus's perceptual quality is improved by 0.139 and 0.118 (in LPIPS), respectively, and the proportion of severely distorted frames is reduced by 89.3% and 91.7%, respectively.

**Weaknesses:**

1.	The experiments and evaluation are not sufficient enough.

a) Only the decoding time is listed in the appendix, but not the encoding time.

b) For measuring the generated video sequences, it is recommended to open source these videos, or show examples frame by frame to prove the stability of the generation.

c) Since Promptus is aimed at low-bitrate video compression, it is recommended to release more subjective comparison results to prove the advantages of the Promptus.

d) The paper only mentions the interpolation of prompt. It's suggested to be compared with applying the interpolation in the pixel domain among video frames in terms of generation performance and complexity to prove the effectiveness of interpolation of prompt.

e) There is also a lack of comparison with other specific state-of-the-art video compression methods. It is recommended to compare with more video compression methods (e.g. [1][2]).

[1] Li J, Li B, Lu Y. Neural video compression with feature modulation[C]//Proceedings of the IEEE/CVF Conference on Computer Vision and Pattern Recognition. 2024: 26099-26108.

[2] Sheng X, Li L, Liu D, et al. Prediction and Reference Quality Adaptation for Learned Video Compression[J]. arXiv preprint arXiv:2406.14118, 2024.

2.	This proposed diffusion-based video compression method shows several limitations. For example, the relatively large delay and complexity, because the prompt needs to be inserted into the intermediate frame, so the subsequent frames must be received before playback. The second is the usage scenario, which is currently limited to low-bitrate video streaming scenarios.

**Questions:**

1.	The comparison with more advanced codec like H266 is also suggested.
2.	Since SD is not very good at generating features like human fingers and text, could you show how Promptus performs in places with a lot of high-frequency information such as fingers and text?

---

> ### Author Response · Authors · 2024-11-24
> **Response to Reviewer W5gp**
>
> Thank you for the valuable feedbacks. A revision version has been uploaded, where modifications are highlighted in red. Below is our response to your questions and concerns. The original comments are copied followed by our answers.
>
> > W1: Only the decoding time is listed in the appendix, but not the encoding time.
>
> Thank you for suggesting this! We added more details about the encoding process in Section C, including time overhead. Additionally, we added subjective examples of encoding results at different iterations from initialization to convergence in Figure 14.
>
> > W2: ...it is recommended to open source these videos, or show examples frame by frame to prove the stability of the generation.
>
> Thank you for raising this insightful question. We added frame-by-frame examples to Figure 15 and second-by-second examples to Figure 11. Additionally, in response to Reviewer qpvP's suggestion, We also added the X-t slice experiment into Figure 11 to evaluate the stability of the generation. The results indicate that our videos basically aligns with the ground truth videos in terms of motion.
>
> Moreover, we will open source the videos and codes after publication.
>
> > W3: it is recommended to release more subjective comparison results...
>
> Thanks for the valuable suggestion. We added 5 subjective experiments from different perspectives: Figure 5 (color discrepancy), Figure 11 (temporal consistency), Figure 14 (training process),  Figure 15 (interpolation experiments and ablation studies）and Figure 16 ("fingers" and "text"). Please refer to the "General response for all reviewers."
>
> > W4: ...It's suggested to be compared with applying the interpolation in the pixel domain among video frames...
>
> Thanks for the helpful suggestion! Pixel interpolation is also effective in reducing the bitrate of videos. We added comparisons of prompt interpolation, pixel interpolation, and latent interpolation in Section D and Figure 15.
>
> For pixel interpolation, we apply RIFE [1], a real-time video frame interpolation algorithm. The results show that it leads to noticeable artifacts in cases of occlusion, edges, or newly appearing objects due to incorrect matching, as illustrated by the green and red boxes in the Figure 15.
>
> For Latent interpolation, it fails to preserve the motion between frames, resulting in spatial overlaps and ghosting, as shown in the second row of the Figure 15.
>
> For our prompt interpolation, it successfully preserves motion while avoiding artifacts and keeping the edges sharp. Additionally, prompt interpolation uses the simplest linear interpolation, which introduces almost no additional overhead.
>
> [1] Zhewei Huang, Tianyuan Zhang, Wen Heng, Boxin Shi, and Shuchang Zhou. Real-time intermediate flow estimation for video frame interpolation. In Proceedings of the European Conference on Computer Vision (ECCV), 2022.
>
> > W5: ...the prompt needs to be inserted into the intermediate frame, so the subsequent frames must be received before playback.
>
> Thank you for raising this concern. Promptus is designed for Video on Demand, such as YouTube and Netflix. In this scenario, the video is transmitted in chunks instead of frame by frame, where each chunk is a small segment of video (e.g., 1 second,  30 frames). Once received, chunks are queued in a buffer (such as 10 seconds) for playback. Therefore, users do not perceive the interpolation because it is always completed before video playback.
>
> > W6: The second is the usage scenario, which is currently limited to low-bitrate video streaming scenarios.
>
> Thank you for the valuable feedback. Promptus works well at all bitrate levels. It can reduce the bitrate to very low levels (e.g., 1/4) while maintaining the same video quality. Considering that network traffic is quite expensive, on-demand video alone generates 8 EB of traffic daily over fixed networks [2]. Therefore, the ability of Promptus to reduce bitrates is valuable.
>
> [2] Sandvine. 2024. 2024 Global Internet Phenomena Report. https://www.sandvine.com/global-internet-phenomena-report-2024.
>
> > Q1: The comparison with more advanced codec like H266 is also suggested.
>
> Thank you for suggesting this! We added a comparison with H.266 in Section 4.3 Compression Efficiency  and Section B Performance on Real-world Traces. The results indicate that Promptus maintains its advantages.
>
> > Q2: ...could you show how Promptus performs in places with a lot of high-frequency information such as fingers and text?
>
> Thanks for raising this insightful concern! We added experiments on "fingers" and "text", as shown in Figure 16. The results show that Promptus can generate "fingers" and "text" quite well. This is because the inherent issues in SD can be compensated for during the end-to-end gradient descent fitting.

---

> > ### Comment · Reviewer_W5gp · 2024-11-26
> > **A few more questions about the response**
> >
> > Thank you for your detailed response and experiments. I have a couple of questions:
> >
> > 1. In Figure 15, the paper mentions that keyframe 1 and keyframe 10 are provided, and intermediate frames are interpolated to compare methods. Keyframe 10 should be the same for all methods, but the ‘Latent Interpolation’ results look much worse. The author may need to explain the reason.
> >
> > 2. In Figure 16, the examples with text and fingers are interesting, but there’s no mention of the bitrate overhead. Could you share more details about this?

---

> > > ### Author Response · Authors · 2024-11-26
> > > **Response to Reviewer W5gp**
> > >
> > > Thank you for your reply and for giving more valuable suggestions. A revision version has been uploaded. Below is our response to your questions.
> > >
> > > > Q1: In Figure 15, the paper mentions that keyframe 1 and keyframe 10 are provided, and intermediate frames are interpolated to compare methods. Keyframe 10 should be the same for all methods, but the ‘Latent Interpolation’ results look much worse. The author may need to explain the reason.
> > >
> > > Thank you for raising this question. This detail you pointed out is very important. Keyframe 10 of "latent interpolation" in Figure 15 looks worse due to the temporal smoothing regularization used during fitting. This is because the experiments on pixel and latent interpolation are end-to-end, rather than based on the intermediate results from Promptus. Therefore, the key frames in pixel interpolation are the ground truth images, while the key frames in latent interpolation are the latent variables fitted using VAE as the generator. Because the frames are not temporally close in the latent space, enforcing a temporal smoothing regularization leads to artifacts in the fitting results.
> > >
> > > We agree with your suggestion that Keyframe 10 should be the same for all methods, so we added this setting in Figure 15 (using images and latent variables from Promptus as keyframes). The results indicate that the conclusions remain the same, because the average performance of the "end-to-end" setting is actually better. We also added a discussion about this in Section D.
> > >
> > > > Q2: In Figure 16, the examples with text and fingers are interesting, but there’s no mention of the bitrate overhead. Could you share more details about this?
> > >
> > > Thank you for pointing that out kindly! The bitrate overhead for the examples in Figure 16 is 225 kbps (with a rank of 8). We will add this to the revised version.
> > >
> > > Thank you again for the discussion. Your insights are valuable to us.

---

> ### Author Response · Authors · 2024-12-01
> **Follow-up**
>
> Dear Reviewer W5gp,
>
> Thank you once again for your dedicated review! As the deadline for the author-reviewer discussion phase approaches, we eagerly await your feedback on our responses. Any insights you provide are greatly appreciated and will help us further improve this work.
>
> Thank you so much!
>
> The authors

---

### Official Review · Reviewer_5rEN · 2024-10-31

**Soundness:** 3
**Presentation:** 3
**Contribution:** 3
**Rating:** 6
**Confidence:** 3

**Summary:**

This paper introduces Promptus, a new video streaming system that reduces bandwidth by transmitting video frames as Stable Diffusion prompts rather than raw video data. Utilizing gradient descent for pixel alignment and low-rank decomposition for bitrate control, Promptus achieves over 4x bandwidth reduction compared to traditional methods, maintaining high perceptual quality at low bitrates.

**Strengths:**

1. Novel in the sense of reforming the paradigm of traditional video streaming system by transmitting prompt streaming instead of video streaming.
2. Ensure pixel alignment in this kind of paradigm based on generative model is inspiring.
3. The workflow of the proposed method is well presented. The effectiveness of components is well discussed.
4. The authors openly discuss limitations of using prompts for video streaming system.

**Weaknesses:**

1. The authors discuss recent SOTA works on neural-based video compression in the “Introduction” and “Related Works” sections. However, Figures 8, 9, and 10 do not include a comparison of the results with representative methods from these works. Additionally, what is the significance of the comparison with VAE?
2. Although the authors address this in the paper, I do think that PSNR and SSIM should still be considered alongside LPIPS.
3. I think the paper should include more comparisons of subjective results at different bitrates to highlight its advantages on pixel alignment.
4. Since the paper introduces Stable Diffusion as part of the method, and the authors have also noted this increase in complexity, can the increase in model parameters be discussed? Furthermore, can more details on the training process be provided?

**Questions:**

The questions are mentioned in the “Weaknesses” section.

What’s more, the citation format needs further adjustments according to the conference requirements. e.g., VP8 (Bankoski et al., 2011) instead of  “VP8 Bankoski et al. (2011)”.

---

> ### Author Response · Authors · 2024-11-24
> **Response to Reviewer 5rEN**
>
> We appreciate the reviewer for the insightful questions. A revision version has been uploaded, where modifications are highlighted in red. Below is our response to your questions and concerns. The original comments are copied followed by our answers.
>
> > W1: ...Figures 8, 9, and 10 do not include a comparison of the results with representative methods from these works...
>
> Thank you for the valuable feedback. We added a comparison with H.266 in Section 4.3 Compression Efficiency and Section B Performance on Real-world Traces. This is a state-of-the-art video codec that is widely compared with other state-of-the-art works, making it a suitable reference. Many of the discussed works in "Related Work" are optimized for specific scenarios and therefore cannot be compared as general compression solutions. For example, Face-vid2vid [1] [2] can only compress videos of human faces in video conferencing. In the future work, we will try to compare with more representative methods.
>
> [1] Ting-Chun Wang, Arun Mallya, and Ming-Yu Liu. One-shot free-view neural talking-head synthesis for video conferencing. In Proceedings of the IEEE/CVF conference on computer vision and pattern recognition, pp. 10039–10049, 2021.
>
> [2] Peiwen Jiang, Chao-Kai Wen, Shi Jin, and Geoffrey Ye Li. Wireless semantic communications for video conferencing. IEEE Journal on Selected Areas in Communications, 41(1):230–244, 2022.
>
> > W2: what is the significance of the comparison with VAE?
>
> Thank you for raising this valuable question. The comparison with VAE serves as an ablation study, as Promptus's Stable Diffusion includes the U-Net and VAE modules. The unsatisfactory performance with only VAE demonstrates that the U-Net module (Diffusion process) also plays a significant role in compression performance. Therefore, VAE and U-Net need to work together to achieve the performance of Promptus.
>
> > W3: I think the paper should include more comparisons of subjective results...
>
> Thanks for the valuable suggestion. We added 5 subjective experiments from different perspectives: Figure 5 (color discrepancy), Figure 11 (temporal consistency), Figure 14 (training process),  Figure 15 (interpolation experiments and ablation studies）and Figure 16 ("fingers" and "text"). Please refer to the "General response for all reviewers."
>
> > W4: can the increase in model parameters be discussed?
>
> Thank you for suggesting this! We added the specific model parameter amounts and some discussions in Section C. With more parameters (such as SD XL Turbo), the SD model has stronger generative ability, making prompt fitting easier and allowing for higher compression rates. However, this also leads to increased overhead in terms of memory usage and run time.
>
> > W5: can more details on the training process be provided?
>
> Thanks for the valuable advice. We added more details about the training process in Section C. Additionally, we added subjective examples of fitting results at different iterations from initialization to convergence in Figure 14.
>
> > W6: PSNR and SSIM should still be considered alongside LPIPS.
>
> Thank you for your valuable suggestion. We use LPIPS because Promptus is more sensitive to sharp textures like all AIGC models, while PSNR and SSIM are more sensitive to smooth textures with large areas. As a result, the images generated by Promptus have sharper details with better perceptual quality, while the baselines are more blurred but with higher PSNR. Sometimes, the more details presented by Promptus, the lower the PSNR, while the more blurred the baselines are, the higher the PSNR can be. This phenomenon is also illustrated in the subjective examples in Figure 10.
>
> We agree that PSNR and SSIM should be considered in the video compression task. However, in this paper, Promptus is more oriented towards the semantic communication task. The main goal is to keep the semantic information correct at low bitrates. LPIPS considers perceptual quality, making it more suitable for evaluating semantic distortion. Promptus is a novel paradigm for high-fidelity semantic communications. We will add PSNR and SSIM in the future work.
>
> > Q1: The citation format needs further adjustments...
>
> Thank you for pointing that out! We have modified the citation format as suggested.

---

> ### Author Response · Authors · 2024-12-01
> **Follow-up**
>
> Dear Reviewer 5rEN,
>
> Thank you once again for your dedicated review! As the deadline for the author-reviewer discussion phase approaches, we eagerly await your feedback on our responses. Any insights you provide are greatly appreciated and will help us further improve this work.
>
> Thank you so much!
>
> The authors

---

### Official Review · Reviewer_pcrw · 2024-11-02

**Soundness:** 3
**Presentation:** 3
**Contribution:** 3
**Rating:** 8
**Confidence:** 4

**Summary:**

The paper proposed an interesting new paradigm for representing videos by inversing them into prompts, replacing traditional video encoding/decoding. In video communication, the sender transmits prompts, while the receiver uses these prompts to control Stable Diffusion to generate (reconstruct) the original video. It is impressive that the generated video can achieve pixel alignment with the original video, and the gradient descent fitting approach is quite pleasing. The methods of controlling bitrate through low-rank decomposition and achieving inter-frame compression via prompt interpolation are sound. The results, particularly in comparison to H.265, demonstrate the potential of Promptus in video communication.

**Strengths:**

1. Clear setting. Reducing video bitrate is a practical and important problem, as video traffic can be quite expensive for content providers. This paper offers a revolutionary new perspective by representing videos with prompts and employing generative models for video reconstruction.

2. Well motivation. The main ideas and tricks can be easily understood. Many concerns at first glance are addressed in the following sections. The paper systematically discusses the challenges of prompt streaming, such as bitrate control, inter-frame compression, and real-time playback, and presents a comprehensive solution.

3. Simple yet effective, making it easy to follow. In the gradient descent framework, there appear to be many potential improvements for further work.

**Weaknesses:**

1. Considering that random packet loss occurs in network transmission, how would this influence the performance of Promptus? Would incomplete prompts severely degrade the quality of the generated video? Please clarify.

2. To reduce bitrate, semantic communication is an emerging paradigm. Although some related work is introduced in the related work section, the authors do not explicitly compare Promptus with semantic communication. Is Promptus an instance of semantic communication or a potential alternative? Please clarify.

3. This paper should thoroughly clarify the fundamental differences between prompt inversion and video encoding/decoding. While P5 L269 touches on this point, the authors do not provide further explanation.

4. What is the scalability of Promptus? Although it can generate videos in real-time from prompts, the overhead of the inversion from video to prompts cannot be ignored, as acknowledged in the limitation section. How would this cost increase on a video platform with billions of users?

**Questions:**

Please address the weaknesses above.

---

> ### Author Response · Authors · 2024-11-24
> **Response to Reviewer pcrw**
>
> We thank the reviewer for the valuable feedbacks. A revision version has been uploaded, where modifications are highlighted in red. Below is our response to your questions and concerns. The original comments are copied followed by our answers.
>
> > W1: Considering that random packet loss occurs in network transmission, how would this influence the performance of Promptus?
>
> Thank you for raising this question. Promptus is used for Video on Demand. Currently, most on-demand videos are transmitted using HTTP, based on the TCP protocol. This means that data transmission is reliable.
>
> In future work, we will extend Promptus to RTC scenarios, where packet loss may occur, necessitating the design of error recovery mechanisms for Promptus.
>
> > W2: Is Promptus an instance of semantic communication or a potential alternative?
>
> Thank you for the insightful feedback. Promptus represents a new paradigm in semantic video communication. It uses prompts for communication, which inherently belong to semantic information. However, traditional semantic communication only aim for semantic consistency and cannot achieve pixel-level consistency, as shown in Figure 1. Promptus further ensures pixel consistency, thereby broadening the application of semantic communication to some high-fidelity scenarios.
>
> > W3: This paper should thoroughly clarify the fundamental differences between prompt inversion and video encoding/decoding.
>
> Thank you for the valuable feedback. Video encoding records the video signal itself. To ensure high fidelity of the signal, the compression rate is limited. While Promptus records the coordinates of the video in the prompt space, instead of the signal itself. This achieves a better compression rate while ensuring fidelity. We added an explanation about this in Section 3.1.
>
> > W4: ...How would this cost increase on a video platform with billions of users?
>
> Thank you for raising this concern. Since Promptus is used for Video on Demand, the videos in this scenario are pre-encoded and stored. Therefore, the cost of encoding (inversion) is a one-time expense, and subsequent use only requires decoding (generation). The generation takes place on the user's device, so the number of users does not affect the costs of Promptus on the video platform side.

---

> > ### Comment · Reviewer_pcrw · 2024-11-25
> >
> > Thanks for your response. All my questions are addressed and I will improve my score.
> >
> > I think Promptus will attract extensive attention in the semantic communication task, because of its pixel consistency. I agree that the application of Promptus in on-demand video streaming is promising. I look forward to seeing its application extended to RTC.

---

> > > ### Author Response · Authors · 2024-11-26
> > > **Thanks for the reply**
> > >
> > > We thank the reviewer for the insightful feedback and for increasing the score!   We will continue to expand Promptus.

---

### Official Review · Reviewer_Jrjx · 2024-11-04

**Soundness:** 3
**Presentation:** 3
**Contribution:** 3
**Rating:** 5
**Confidence:** 4

**Summary:**

The paper introduces Promptus, an innovative system that represents real-world video frames as prompts for Stable Diffusion, enabling ultra-low bitrate video streaming by sending prompts instead of raw video content. Evaluations show that Promptus achieves more than a 4x reduction in bandwidth compared to H.265 while preserving similar perceptual quality and demonstrating significant quality improvements at extremely low bitrates.

**Strengths:**

1. Novel application of GAN AI for video streaming: By leveraging Stable Diffusion for video generation through prompts, Promptus introduces a novel approach to reducing video bandwidth requirements.
2. The gradient descent-based prompt fitting framework ensures pixel-level alignment, maintaining visual consistency with original frames, which is challenging for generative models.
3. The low-rank decomposition approach allows Promptus to adjust bitrate based on network conditions, enhancing usability in fluctuating bandwidth situation.
4. Temporal smoothing regularization reduces the need to transmit prompts for every frame, significantly cutting down bandwidth usage.
5. Promptus shows a marked improvement in bandwidth efficiency, achieving a fourfold reduction over H.265 while maintaining perceptual quality, especially in complex or high-frequency video content.

**Weaknesses:**

1. Promptus’s primary goal is to find an inverse prompt that moves points represented by noise to the target image’s location in latent space. However, it’s unclear how the model handles abrupt scene cuts within videos. Scene cuts could disrupt continuity, as prompts optimized for one scene may not generalize to an entirely different visual context, the generated frames might misrepresent scene transitions.
2. As described in 485 lines, as bitrate decreases, Promptus reduces the descriptive capacity of prompts. How the model maintains accurate content representation at these lower bitrates. Why this simplification not lead to slight misalignments in generated frames? More insights into the model’s mechanisms for preserving spatial and temporal consistency at low bitrates would clarify this point.
3. The paper predominantly relies on LPIPS as the primary evaluation metric. Incorporating additional metrics, such as SSIM or VMAF, would provide a more holistic assessment of visual quality, particularly in capturing structural integrity and perceptual alignment.
4. Although the supplementary materials provide information on the complexity of the proposed method, the paper lacks direct comparisons with the computational complexity and runtime of other state-of-the-art (SOTA) benchmarks.
5. The experimental results compare Promptus only with H.265 and a VAE model, omitting comparisons with other recent SOTA benchmarks in video compression and generation.

**Questions:**

1. The paper primarily uses linear interpolation for temporal smoothing between frames. Has the team explored other interpolation methods?
2. The visualizations in Figure 10 show noticeable color discrepancies between the generated and original videos. Was the model trained in RGB or YUV color space?
3. Figure 5 shows that while fine details such as hair strands are preserved, specific elements like earrings disappear. Why does Promptus struggle to preserve certain details?

---

> ### Author Response · Authors · 2024-11-24
> **Response to Reviewer Jrjx**
>
> We thank the reviewer for the valuable feedbacks. A revision version has been uploaded, where modifications are highlighted in red. Below is our response to your questions and concerns. The original comments are copied followed by our answers.
>
> > W1: ...it’s unclear how the model handles abrupt scene cuts within videos...
>
> We agree with your concerns, as adjacent frames in the prompt space are no longer close after the scene changes, making interpolation not work. To address this, Promptus will continuously detect scene changes and treat the new scenes as new videos, as described in Section A.1. To highlight this, we added a reference to this part in Section 3.3 of the main text.
>
> > W2: ...How the model maintains accurate content representation at these lower bitrates. Why this simplification not lead to slight misalignments in generated frames?...
>
> Thank you for raising this question. Low-bitrate prompts are also obtained through end-to-end gradient descent fitting, ensuring that the generated images are as consistent as possible with the ground truth images. Thus, Promptus makes the most of the bitrate, achieving the best consistency at low bitrates.
>
> > W4: ...lacks direct comparisons with the computational complexity and runtime of other state-of-the-art (SOTA) benchmarks...
>
> Thanks for the valuable advice. According to Table 1, during decoding, Promptus introduces almost no additional overhead (only some simple linear computations), with the majority of the overhead coming from SD itself. As a pipeline, Promptus can be compatible with different SD models. We believe that with the development of the SD community, more lightweight SD models will emerge, benefiting Promptus. In future work, we will evaluate complexity and runtime on more SOTA benchmarks.
>
> > W5: ...omitting comparisons with other recent SOTA benchmarks in video compression and generation...
>
> Thanks for the suggestions! We added a comparison with H.266 in Section 4.3 Compression Efficiency and Section B Performance on Real-world Traces. This is a state-of-the-art video codec that is widely compared with other state-of-the-art works, making it a suitable reference.
>
> > Q1: Has the team explored other interpolation methods?
>
> Thanks for the insightful advice. We added subjective experiments on different interpolation methods in Section D and Figure 15.
>
> For prompt interpolation, we explored other one-dimensional interpolation methods, such as cubic interpolation, but they did not differ much from linear interpolation. To maintain the simplicity, we finally chose linear interpolation.
>
> We further compared interpolation at the prompt level, latent level, and pixel level (in response to Reviewer W5gp's suggestions). The results demonstrated the superiority of prompt interpolation, as shown in Figure 15.
>
> > Q2: Figure 10 show noticeable color discrepancies between the generated and original videos. Was the model trained in RGB or YUV color space?
>
> Thank you for raising this concern. Promptus is trained in the RGB space. However, the color discrepancies in Figure 10 primarily stem from the low bitrate of the prompt. We added subjective examples to Figure 5, which shows that when the rank (bitrate) is low, the lamp’s color in Figure 5(d) is inconsistent with the ground truth. When the rank increases, the lamp’s color in Figure 5(e) is corrected. This is because when the bitrate is low, the representational capacity of the prompt decreases, making it unable to accurately describe all the details in the image, resulting in color discrepancies.
>
> > Q3: Why does Promptus struggle to preserve certain details?
>
> It is an interesting question. This is because SD itself has varying abilities for generating different elements. For specific elements, such as text, SD itself struggles to produce them. Fortunately, Promptus can successfully fit these elements through end-to-end gradient descent, as shown in Figures 5 and 16, although this requires more iterations.

---

> > ### Comment · Reviewer_Jrjx · 2024-11-26
> > **A few more questions about the response**
> >
> > Thank you for your detailed responses to my comments and for incorporating additional experiments, especially the comparison with H.266. I appreciate your effort to address the points raised and enhance the manuscript. However, I believe some concerns remain insufficiently addressed:
> >
> > Evaluation Metrics: While I understand that LPIPS provides insight into perceptual quality, for video compression, the most widely recognized metrics are PSNR and VMAF, which are critical for evaluating structural integrity and perceptual alignment. I encourage you to include these metrics to provide a more holistic assessment of your method's performance and to align with the standard practices in the field.
> >
> > Details of H.266 Comparison: In the newly added comparisons with H.266, the paper lacks sufficient detail regarding the codec model and compression parameters used. For the results to be interpretable and reproducible, it is essential to specify whether VTM and its version or another implementation was used, as well as the precise settings (e.g., GOP size, CRF, or bitrate). Without this information, it remains unclear how robust the comparisons are.
> >
> > Overall, while I appreciate the improvements made to the manuscript. However, I feel the authors have not directly addressed my concerns about using the evaluation metrics, avoiding discussion of their method's performance on these benchmarks. This lack of engagement with standard metrics leads me to reconsider my scoring for the manuscript.

---

> > > ### Author Response · Authors · 2024-11-27
> > > **Response to Reviewer Jrjx**
> > >
> > > Thank you for your reply. Below is our response to your questions.
> > >
> > > > Q1: Evaluation Metrics
> > >
> > > Thanks for your comments. Your suggestion is valuable. We tested the performance of PSNR and SSIM on UVG, with a bitrate of 225 kbps. The results are as follows:
> > >
> > > | Metrics   | H.266   | H.265   | Ours   |
> > > |-------|-------|-------|-------|
> > > | psnr   | 34.14 | 32.60 | 30.04 |
> > > | ssim   | 0.80 | 0.72 | 0.61 |
> > > | lpips   | 0.32 | 0.39 | 0.24 |
> > >
> > > This is because Promptus is more sensitive to sharp textures like all AIGC models, while PSNR and SSIM are more sensitive to smooth textures with large areas. As a result, the images generated by Promptus have sharper details with better perceptual quality, while the baselines are more blurred but with higher PSNR. Sometimes, the more details presented by Promptus, the lower the PSNR, while the more blurred the baselines are, the higher the PSNR can be. This phenomenon is also illustrated in the subjective examples in Figure 10.
> > >
> > > We agree that PSNR and SSIM are widely used in the video compression task. However, in this paper, Promptus is more oriented towards the semantic communication task. The main goal is to keep the semantic information correct at low bitrates. LPIPS considers perceptual quality, making it more suitable for evaluating semantic distortion. Promptus is a novel paradigm for high-fidelity semantic communications. There are still issues that need to be fixed in future work.
> > >
> > >
> > > > Q2: Details of H.266 Comparison
> > >
> > > Thank you for pointing that out! The encoder implementation of H.266 is [1], version 1.12.1-rc1, and the decoder implementation is [2], version 3.0.0. As for the encoding settings, we specified only the resolution (512x512), target bitrates (140 kbps, 280 kbps, 360 kbps, 540 kbps), and frame rate (30 FPS). All other settings were kept at default, and we did not specify them.
> > >
> > > [1] VVenC. https://github.com/fraunhoferhhi/vvenc.
> > >
> > > [2] VVdeC. https://github.com/fraunhoferhhi/vvdec.
> > >
> > > For reproducibility, we will open-source the encoded videos and source code after publication.
> > >
> > > Thank you for the discussion and look forward to your reply. We are making the paper more solid together.

---

> ### Author Response · Authors · 2024-12-01
> **Follow-up**
>
> Dear Reviewer Jrjx,
>
> Thank you once again for your dedicated review! As the deadline for the author-reviewer discussion phase approaches, we eagerly await your feedback on our responses. Any insights you provide are greatly appreciated and will help us further improve this work.
>
> Thank you so much!
>
> The authors

---

### Official Review · Reviewer_LCsu · 2024-11-04

**Soundness:** 2
**Presentation:** 2
**Contribution:** 2
**Rating:** 5
**Confidence:** 4

**Summary:**

The paper proposed a pipeline called Promptus, which uses prompts instead of video content for streaming and uses Stream Diffusion to generate video at the receiving end. In order to ensure that the prompt representation is aligned with the original video, a prompt fitting framework based on gradient descent is proposed. In addition, a bitrate control algorithm based on low-rank decomposition is introduced to achieve adaptive bitrate. The paper conducted experiments on several individual videos such as QSR Animerun to prove that Promptus can achieve more than 4 times bandwidth reduction while maintaining the same perceptual quality. At the same time, the perceptual quality is higher than that of traditional methods at extremely low bitrates.

**Strengths:**

Modules such as gradient descent based prompt fitting are proposed to solve the inconsistency problem of diffusion model generation in video streaming transmission.
Experiments demonstrate the effectiveness of this method in LPIPS, compression ratio.
The real-time issues are well covered in the supplementary material.

**Weaknesses:**

According to the supplementary, Promptus requires 8952MB of memory to run, but most existing methods [1, 2] do not require such a high video memory requirement, and in actual scenarios, many devices do not have such a high video memory, such as mobile phones. Therefore, this diffusion model-based method can only be used on high-performance PCs with high video memory in the context that diffusion models still require high memory.

As far as I know, there are still many unique problems in the diffusion model in content generation, such as "multiple fingers" and "words cannot express" problems. I hope the author can provide relevant experiments to prove what results the method in this paper will have when facing the unique problems of SD itself.

Although the author said in the experiment that the resolution can be arbitrary, since this work relies on the representation capability of SreamDiffusion[3] based on SD-Turbo or LCM, many "high-resolution" images are not included in the training set of SD-Turbo or LCM. Can Promptus effectively transmit videos with various resolutions and high resolutions that it has not seen during SD training?

The specific diffusion model version used is not written in the main text. It is recommended to write it in the main text. In addition, the time speed and memory overhead of this method will be the primary consideration for most people. I suggest the author will move such experiments into the main paper.

[1] H.265,2024.https://www.itu.int/rec/T-REC-H.265
[2] Gemino: Practicalandrobustneural compressionforvideoconferencing.
[3] https://github.com/cumulo-autumn/StreamDiffusion

**Questions:**

This paper is novel in the field of video streaming, and the idea of using prompt to transmit video is very interesting. However, the application scenarios are narrow, and there is a lack of more feasibility experiments on diffusion models. Therefore, I suggest that the author first provide sufficient evidence to fully demonstrate that Stable Diffusion is a feasible representation.

---

> ### Author Response · Authors · 2024-11-24
> **Response to Reviewer LCsu**
>
> We appreciate the reviewer for the insightful questions. A revision version has been uploaded, where modifications are highlighted in red. Below is our response to your questions and concerns. The original comments are copied followed by our answers.
>
> > W1: ...this diffusion model-based method can only be used on high-performance PCs with high video memory...
>
> Your concern about memory overhead is important, considering the significant memory usage of current Stable Diffusion. We have made efforts to decrease the memory usage of Promptus. For example, we replaced the Stable Diffusion decoder (49.5M parameters) with a more lightweight TAESD Decoder (1.2 M parameters). Furthermore, as a pipeline, Promptus can be compatible with different SD models. We believe that with the development of the SD community, more lightweight SD models will emerge, enabling Promptus to run on some mobile devices. In our follow-up work, we will also focus on reducing memory usage.
>
> > W2: ...there are still many unique problems in the diffusion model in content generation, such as "multiple fingers" and "words cannot express" problems. I hope the author can provide relevant experiments...
>
> Thanks for raising this concern! We added experiments on "fingers" and "text", as shown in Figure 16. The results show that Promptus can generate "fingers" and "text" quite well. This is because the inherent issues in SD can be compensated for during the end-to-end gradient descent prompt fitting.
>
> > W3: ...Promptus effectively transmit videos with various resolutions and high resolutions that it has not seen during SD training?
>
> Thank you for your insightful suggestion. The SD model does struggle to generate good images for resolutions it hasn't seen during training. However, thanks to the end-to-end gradient descent fitting, Promptus can generate good images for these challenging resolutions, just like the example about "fingers" mentioned above. In our future work, we will include more experiments at higher resolutions.
>
> >  W4: The specific diffusion model version used is not written in the main text.
>
> Thank you so much for pointing this out！We added in Section 3.1 that the version is SD 2.1 Turbo.
>
> > W5: ...the time speed and memory overhead of this method will be the primary consideration for most people. I suggest the author will move such experiments into the main paper.
>
> Thanks for the valuable advice. We moved Table 1 (overhead results) to the main text. Due to space limit, more results and analysis are left in Section C. So we added references to Section C in the main text.
>
> > Q1: This paper is novel in the field of video streaming, and the idea of using prompt to transmit video is very interesting. However, the application scenarios are narrow, I suggest that the author first provide sufficient evidence to fully demonstrate that Stable Diffusion is a feasible representation.
>
> Thank you for the feedback. We added 7 experiments from different perspectives; please refer to the "General response for all reviewers." The results demonstrate that Promptus can generate challenging elements such as "fingers", support high resolutions, and align with the ground truth video in motion. We have made efforts to reduce memory usage and will continue to improve this in the future work.
>
> The current version of Promptus can be used for Video on Demand on PCs equipped with high-performance GPUs. This application scenario holds significant value. According to the 2024 Global Internet Phenomena Report [1], excluding mobile networks, there are currently 1.4 billion fixed network users, each consuming an average of 5.7 GB of Internet traffic daily on Video on Demand (such as YouTube and Netflix), totaling 1.4 billion * 5.7 GB = 8 EB of traffic, which accounts for the largest portion of Internet traffic volume (39%). Considering that traffic is quite expensive, Promptus's ability to reduce the bitrate by 4x is valuable.
>
> In the future, we will continue to improve Promptus to support live video, real-time communication (RTC), and enable it to run on mobile devices.
>
> [1]Sandvine. 2024. 2024 Global Internet Phenomena Report. https://www.sandvine.com/global-internet-phenomena-report-2024.

---

> ### Author Response · Authors · 2024-12-01
> **Follow-up**
>
> Dear Reviewer LCsu,
>
> Thank you once again for your dedicated review! As the deadline for the author-reviewer discussion phase approaches, we eagerly await your feedback on our responses. Any insights you provide are greatly appreciated and will help us further improve this work.
>
> Thank you so much!
>
> The authors

---

### Official Review · Reviewer_qpvP · 2024-11-05

**Soundness:** 2
**Presentation:** 3
**Contribution:** 2
**Rating:** 5
**Confidence:** 4

**Summary:**

In this paper, the authors leverage a stable diffusion model as a compression method to encode video into text (embedding format), using this text embedding as the video representation. The entire process resembles information distillation, where trainable low-rank features are used to form the prompt embedding. After optimization, these low-rank factor matrices become the compressed video representation. The authors propose several solutions to control bitrate, perform inter-frame compression, and ensure pixel alignment.

**Strengths:**

(+) Using stable diffusion as a distillation approach is interesting.

(+) The paper is well-written, with most figures well-designed, illustrated, and easy to follow.

**Weaknesses:**

(-) My major concern is whether there are severe flicker issues or temporal over-smoothing in the decoded video, as the authors did not submit any video as a supplementary file. Considering that the rebuttal can only show images, I suggest the authors present an x-t slice, as used in [1], for a highly dynamic video. The x-t slice would provide a better visual understanding on the temporal consistency.

(-) Could the authors justify why it is necessary to use stable diffusion as the intermediate medium for video compression? Why not directly use CLIP or the SD decoder to distill the frames? Is it because CLIP is not powerful enough?

(-) The VAE could introduce significant color tone-mapping issues and spatial blurriness. Did the authors encounter similar issues on a large scale? I can see severe color issues in Figure 10, particularly in the boy's eye and background. Consequently, I am also concerned about some analyses in Section 4.4. In my understanding, the performance drop on the Animerun dataset occurs because those frames are edge cases for SD/VAE, which is why there are numerous color mapping issues.

(-) Since the VAE and diffusion models are fixed, how do the authors ensure that the frames being represented and compressed fall within the diffusion model's domain of knowledge? Additionally, is the main source of encoding performance derived from CLIP/VAE or the U-Net? I did not see a related discussion on this matter.

(-) The qualitative result comparisons are insufficient and do not convincingly demonstrate the proposed method's promise.

[1] Li, Zhengqi, et al. "Generative image dynamics." Proceedings of the IEEE/CVF Conference on Computer Vision and Pattern Recognition. 2024.

**Questions:**

All my questions are outlined in the weaknesses section. My major concerns are the necessity of using stable diffusion as a streaming approach, the insufficient qualitative result comparisons, and the generalization issues related to the VAE or other components.

---

> ### Author Response · Authors · 2024-11-24
> **Response to Reviewer qpvP**
>
> Thank you for the valuable feedback. A revision version has been uploaded, where modifications are highlighted in red. Below is our response to your questions and concerns. The original comments are copied followed by our answers.
>
> > W1: ...I suggest the authors present an x-t slice, as used in [1], for a highly dynamic video. The x-t slice would provide a better visual understanding on the temporal consistency.
>
> Thank you for your valuable suggestions! We agree that temporal consistency is crucial for the generation of Stable Diffusion. Following your suggestions, we added the X-t slice experiment into Figure 11, where Figure 11(a) represents a high-dynamic video (time-lapse). We also added Figure 15, which shows the video frame by frame (key frames and interpolated frames). The results indicate that our videos basically align with the ground truth videos in terms of motion. This is because Promptus is pixel-consistent with each frame of the ground truth video, ensuring that the temporal consistency also aligns the ground truth video. Additionally, prompt interpolation guarantees the continuity of adjacent frames.
>
> > W2: ...why it is necessary to use stable diffusion as the intermediate medium for video compression? Why not directly use CLIP or the SD decoder to distill the frames?
>
> Thank you for raising this question. The question of which model should serve as the intermediate medium is quite valuable. As for CLIP, it does not have a decoder, which means it can only encode images but cannot generate images.
>
> Using the SD decoder (VAE decoder) as the intermediate medium is a good suggestion, as it can effectively distill frames. However, it cannot perform temporal interpolation. We added distillation experiments where the SD decoder serves as the intermediate medium and presented the interpolation results in Figure 15 (latent interpolation). The results demonstrate that latent interpolation fails to preserve the motion between frames, resulting in spatial overlaps and ghosting. This is because the frames in latent space are not temporally close, making the interpolation unreasonable. To achieve inter-frame compression, one feasible solution is to encode the latent frames using a codec (such as H.265). However, as shown in Figure 9 and Figure 8, this solution performs worse than Promptus, due to the errors introduced by the codec in the latent space.
>
> > W3: The VAE could introduce significant color tone-mapping issues...
>
> Your concern is insightful and we agree that the VAE itself may introduce issues such as color tone mapping. However, since Promptus employs gradient descent to fit the ground truth frames, this issue can be compensated for in end-to-end fitting. Actually, the color discrepancies in Figure 10 are primarily stem from the low bitrate of the prompt. We added subjective examples to Figure 5, which shows that when the rank (bitrate) is low, the lamp’s color in Figure 5(d) is inconsistent with the ground truth. When the rank increases, the lamp’s color in Figure 5(e) is corrected. This is because when the bitrate is low, the representational capacity of the prompt decreases, making it unable to accurately describe all the details in the image, resulting in inconsistent colors.
>
> > W4: ... is the main source of encoding performance derived from CLIP/VAE or the U-Net? ...
>
> Thank you for raising this insightful question. The source of encoding performance is derived from both VAE and the U-Net.
>
> For VAE, it transforms frames from pixel space to latent space, significantly reducing the data size (e.g., 512*512*3 -> 64*64*4，reduced to 1/48). However, VAE cannot perform inter-frame compression, as discussed in the aforementioned W2.
>
> For U-Net, its compression performance comes from two factors. First, by transforming the latent space into the prompt space, it further reduces the data size (e.g., 64*64*4 -> (1024+77)*8, reduced to 1/2). Second, because the frames are temporally close in the prompt space, this enables interpolation-based inter-frame compression, significantly reducing the data size. For example, when the keyframe interval for interpolation is 10, the bitrate reduces to 1/10. In total, U-Net can further compress the data to 1/20 in addition to VAE. We added discussions on this in Section 3.1 and Section D.
>
> In summary, these results demonstrate that VAE and U-Net need to work together to achieve the performance of Promptus.
>
> > W5: The qualitative result comparisons are insufficient.
>
> Thank you for pointing that out. To enrich the qualitative result comparisons, we added 7 experiments from different perspectives: Figure 5 (color discrepancy), Figure 11 (temporal consistency), Figure 14 (training process),  Figure 15 (interpolation experiments and ablation studies), Figure 16 ("fingers" and "text"), Figure 9 (add H.266), Figure 13 (add H.266). Please refer to the "General response for all reviewers."

---

> ### Author Response · Authors · 2024-12-01
> **Follow-up**
>
> Dear Reviewer qpvP,
>
> Thank you once again for your dedicated review! As the deadline for the author-reviewer discussion phase approaches, we eagerly await your feedback on our responses. Any insights you provide are greatly appreciated and will help us further improve this work.
>
> Thank you so much!
>
> The authors

---

### Author Response · Authors · 2024-11-24
**General response for all reviewers**

Dear reviewers,

We appreciate the time you took to review our paper. Your insightful feedbacks have helped us make the paper more solid. We have worked diligently to address your valuable comments, and we hope that the revised paper meets your requirements for publication. Thank you for the opportunity to revise and resubmit our paper. We have uploaded a revised version to OpenReview.

The major modifications are summarized below:

1、We added subjective experiments on "fingers" and "text" in Figure 16. The results show that Promptus can generate them quite well, demonstrating the inherent issues in SD can be compensated for through the end-to-end gradient descent fitting. (In response to reviewer LCsu and W5gp)

2、We added the X-t slice experiment and showed second-by-second video examples into Figure 11 to evaluate the temporal consistency. The results indicate that our videos basically align with the ground truth videos in terms of motion. (In response to reviewer qpvP and W5gp)

3、We added a comparison with H.266 in Section 4.3 Compression Efficiency and Section B Performance on Real-world Traces. The results demonstrate the superiority of Promptus in compression. (In response to reviewer Jrjx, 5rEN and W5gp)

4、We added subjective examples to Figure 5 to prove the color discrepancies are primarily stem from the low bitrate of the prompt, rather than VAE issues. The results show that the color discrepancies disappear as the bitrate increases. (In response to reviewer qpvP and Jrjx）

5、We added ablation experiments where the only the SD (VAE) decoder serves as the generator for inversion in Section D. We also presented the latent interpolation results in Figure 15. The results prove that the U-Net (Diffusion process) plays a significant role in compression performance. (In response to reviewer qpvP, Jrjx and 5rEN)

6、We added pixel interpolation experiments in in Section D and Figure 15. The results demonstrate the superiority of prompt interpolation. (In response to reviewer Jrjx and W5gp)

7、We added more details about the training process in Section C, including overhead. Additionally, we added subjective examples of training results at different iterations from initialization to convergence in Figure 14.  (In response to reviewer 5rEN and W5gp)

Modifications in the revised paper are highlighted in red. Thank you again for your insightful feedbacks.

Best Regards,

Authors.

---

### Comment · Area_Chair_6n6F · 2024-11-25

Hi Reviewers,

We are approaching the deadline for author-reviewer discussion phase. Authors has already provided their rebuttal. In case you haven't checked them, please look at them ASAP. Thanks a million for your help!

---

### Note · Authors · 2025-01-23

**Comment:**

This paper is open source at: https://github.com/JiangkaiWu/Promptus. Welcome to give it a try.

**Withdrawal Confirmation:**

I have read and agree with the venue's withdrawal policy on behalf of myself and my co-authors.